# Identification of a bilirubin receptor that may mediate a component of cholestatic itch

**James Meixiong[1†], Chirag Vasavda[1†], Dustin Green[1], Qin Zheng[1], Lijun Qi[1], Shawn G Kwatra[2], James P Hamilton[3], Solomon H Snyder[1,4,5]\*, Xinzhong Dong[1,2,6,7]\***

[1]The Solomon H. Snyder Department of Neuroscience, Johns Hopkins University School of Medicine, Baltimore, United States; [2]Department of Dermatology, Johns Hopkins University School of Medicine, Baltimore, United States; [3]Division of Gastroenterology and Hepatology, Department of Medicine, Johns Hopkins University School of Medicine, Baltimore, United States; [4]Department of Pharmacology and Molecular Sciences, Johns Hopkins University School of Medicine, Baltimore, United States; [5]Department of Psychiatry and Behavioral Sciences, Johns Hopkins University School of Medicine, Baltimore, United States; [6]Department of Neurosurgery, Johns Hopkins University School of Medicine, Baltimore, United States; [7]Howard Hughes Medical Institute, Johns Hopkins University School of Medicine, Baltimore, United States

**\*For correspondence:**
ssnyder1@jhmi.edu (SHS);
xdong2@jhmi.edu (XD)

[†]These authors contributed equally to this work

**Abstract** Various pathologic conditions result in jaundice, a yellowing of the skin due to a buildup of bilirubin. Patients with jaundice commonly report experiencing an intense non-histaminergic itch. Despite this association, the pruritogenic capacity of bilirubin itself has not been described, and no bilirubin receptor has been identified. Here, we demonstrate that pathophysiologic levels of bilirubin excite peripheral itch sensory neurons and elicit pruritus through MRGPRs, a family of G-protein coupled receptors expressed in primary sensory neurons. Bilirubin binds and activates two MRGPRs, mouse MRGPRA1 and human MRGPRX4. In two mouse models of pathologic hyperbilirubinemia, we show that genetic deletion of either *Mrgpra1* or *Blvra*, the gene that encodes the bilirubin-producing enzyme biliverdin reductase, attenuates itch. Similarly, plasma isolated from hyperbilirubinemic patients evoked itch in wild-type animals but not *Mrgpra1[-/-]* animals. Removing bilirubin decreased the pruritogenic capacity of patient plasma. Based on these data, targeting MRGPRs is a promising strategy for alleviating jaundice-associated itch.
DOI: https://doi.org/10.7554/eLife.44116.001

## Introduction

Chronic pruritus, or itch, is a complex and often debilitating symptom that accompanies a range of cutaneous and non-cutaneous diseases (*Ständer et al., 2007*; *Yosipovitch and Bernhard, 2013a*). The most widely known pruritogen is histamine, which is secreted by mast cells in the skin and activates histamine receptors on nearby sensory neurons (*Bautista et al., 2014*; *Ikoma et al., 2006*; *LaMotte et al., 2014*; *Yosipovitch and Bernhard, 2013b*). While viable treatments exist for histamine-mediated itch, most non-histaminergic conditions are more difficult to treat because the mediators are often unknown (*Kremer et al., 2011*).

**eLife digest** Jaundice causes the skin to yellow as a result of a build-up of a pigment called bilirubin. Normally, bilirubin is made in the liver and removed from the body in digestive fluid called bile, but people with liver or gallbladder problems may end up with too much bilirubin that accumulates in their blood and skin. One side effect of jaundice is intense and uncontrollable itching. Researchers are not sure what causes this itching, and there are few treatments that help to relieve it.

At the molecular level, itching sensations occur when compounds bind to particular receptors on the surface of nerve cells. One family of receptors that can trigger itch is called the Mas-related G-protein Coupled Receptor (MRGPR). Could one of these receptors trigger jaundice-related itching?

Now, Meixiong, Vasavda et al. show that bilirubin binds to and activates MRGPRs to cause itch in mice. Whereas injecting bilirubin into normal mice causes them to scratch, mice that have been genetically engineered to lack MRGPRs do not itch when their own bilirubin levels rise, or when they are injected with bilirubin or with plasma from patients who experience jaundice-related itching. Furthermore, removing bilirubin from the plasma of patients before it was injected into normal mice reduced the amount of itching that the mice felt.

Overall, the results reported by Meixiong, Vasavda et al. suggest that drugs that prevent bilirubin from attaching to MRGPRs might help to alleviate jaundice-related itching. However, researchers must first verify that bilirubin interacts with MRGPRs in people to cause itch. If bilirubin causes itch in people like in mice, scientists could then evaluate existing drugs or make new ones to prevent bilirubin from attaching to the MRGPRs.
DOI: https://doi.org/10.7554/eLife.44116.002

Jaundice, or yellowing of the skin, sclera, and mucosa due to abnormal accumulation of the yellow heme metabolite bilirubin, is commonly associated with chronic non-histaminergic pruritus. Jaundice often presents in patients with hepatobiliary disorders such as cholestasis, characterized by impaired bile flow. Physiologically, bilirubin is typically bound to albumin in serum and concentrates in the liver, where it is conjugated to glucuronic acid and subsequently excreted in bile. At physiologic and mildly elevated concentrations (0.2–2.7 mg/dL, 3.4–46.2 μM), bilirubin is benign. At highly elevated levels however, such as in cutaneous jaundice (>5 mg/dL, >85.5 μM bilirubin), it is associated with pruritus, a correlation first noted by physicians as early as the second century B.C.E. (*Bassari and Koea, 2015*).

## Results

Despite the long-standing association between jaundice and pruritus (*Talwalkar et al., 2003*), bilirubin itself has not been described as a pruritogen. To determine whether bilirubin directly elicits pruritus, we subcutaneously injected bilirubin into the napes of mice. Pathophysiologic concentrations of bilirubin stimulated scratching in a dose-dependent manner at the site of injection (*Figure 1A*). Pre-incubating bilirubin with excess human serum albumin, which binds bilirubin with high affinity (*Breaven et al., 1973*; *Griffiths et al., 1975*; *Jacobsen and Brodersen, 1983*), elicited fewer scratches (*Figure 1A*). The behavioral profile of bilirubin-induced scratching mirrored that of two well-characterized pruritogens, histamine and chloroquine (*Figure 1B*). Notably, histamine and chloroquine only elicit itch when injected into mice at millimolar concentrations despite having nanomolar affinity towards their receptors. In comparison, bilirubin elicited a similar degree of itch even when injected at lower concentrations than histamine or chloroquine (*Figure 1B*). Since mice indiscriminately scratch at the nape if injected with substances that trigger either itch or pain, we also injected mice at the cheek. Unlike at the nape, painful sensations at the cheek evoke a distinct wiping behavior instead of scratching, whereas itchy sensations still elicit scratching (*Shimada and LaMotte, 2008*). Injecting bilirubin in the cheek prompted dose-dependent scratching just as it did at the nape (*Figure 1—figure supplement 1A*). Bilirubin elicited neither wiping nor licking, indicating that it selectively triggers itch and not pain (*Figure 1—figure supplement 1B–C*).

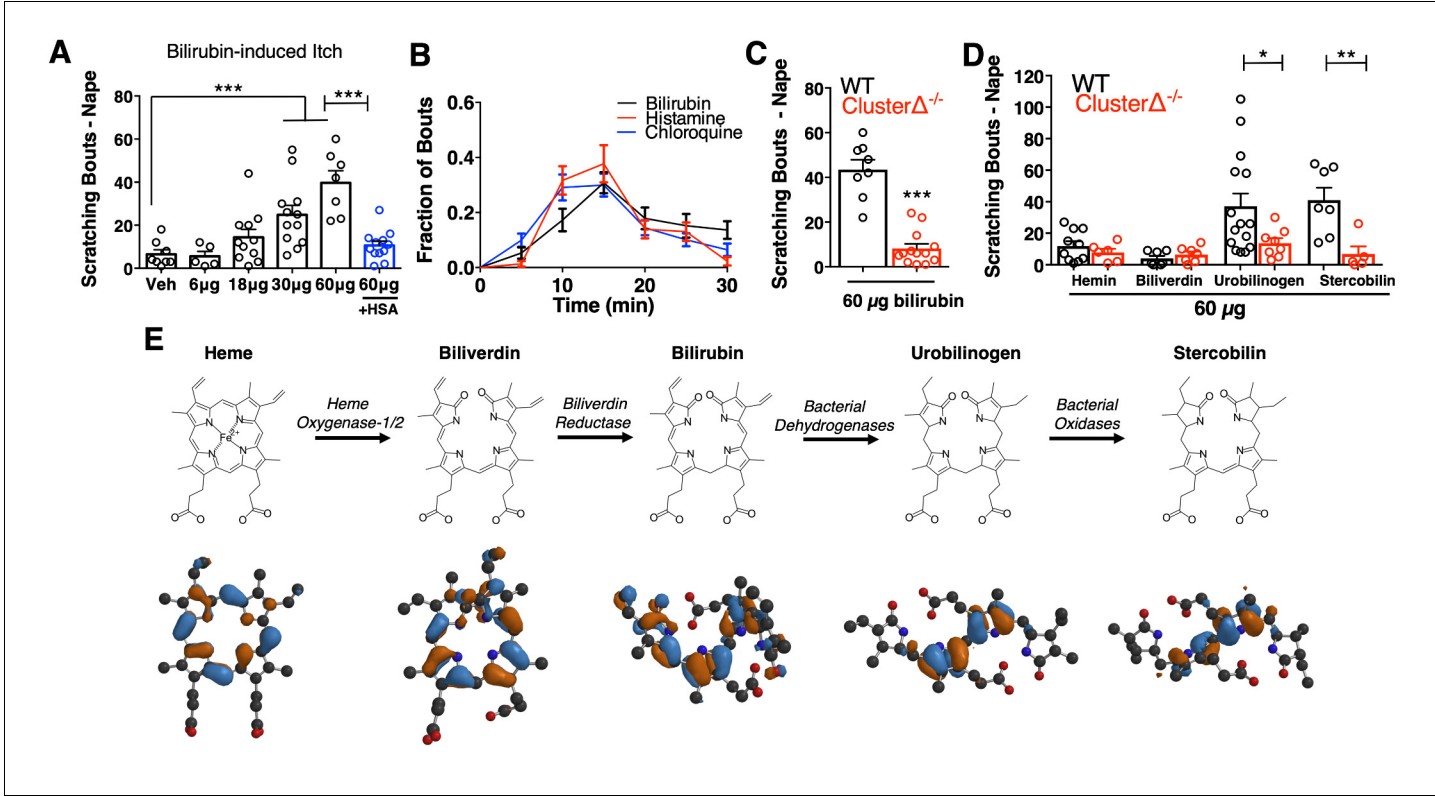

**Figure 1.** Bilirubin elicits non-histaminergic, *Mrgpr*-dependent pruritus. (A) Scratching bouts associated with injection of bilirubin. The indicated amount of bilirubin was injected into the nape of mice. The blue bar (+HSA) represents animals injected with 60 μg bilirubin pre-incubated with 1% human serum albumin. Veh n = 8; 6 μg n = 5, 18 μg n = 11, 30 μg n = 12, 60 μg n = 7,+HSA n = 12. (B) Time course of itch behavior associated with bilirubin, histamine, or chloroquine. Scratching bouts were binned according to 5 min intervals. Bilirubin n = 16, Histamine n = 13, Chloroquine n = 11. (C) 60 μg bilirubin was injected into the nape of WT and Mrgpr-cluster KO littermates. WT n = 8, Mrgpr-cluster KO n = 13. (D) 60 μg of the indicated metabolite was injected into WT and Mrgpr-cluster KO littermates. Hemin (WT n = 10, Mrgpr-cluster KO n = 6), Biliverdin (WT n = 7, Mrgpr-cluster KO n = 7), Urobilinogen (WT n = 15, Mrgpr-cluster KO n = 8), Stercobilin (WT n = 7, Mrgpr-cluster KO n = 5). (E) The pathway of heme degradation. The skeletal formula of each metabolite is depicted above its optimal 3D geometry, as calculated by a B3LYP functional and 6-31G(d) basis set. Blue and orange represent orbital parity of each metabolite's HOMO obtained from DFT calculations. (A, C, D) Mean ±s.e.m. depicted. Each open circle represents an individual mouse. *, $p < 0.05$; **, $p < 0.01$; ***, $p < 0.001$; two-tailed unpaired Student's *t*-test.

DOI: https://doi.org/10.7554/eLife.44116.003
The following source data and figure supplement are available for figure 1:

**Source data 1.** Source data for *Figure 1*.
DOI: https://doi.org/10.7554/eLife.44116.005
**Figure supplement 1.** Bilirubin elicits non-histaminergic pruritus and not pain.
DOI: https://doi.org/10.7554/eLife.44116.004

We injected mice with metabolites structurally similar to bilirubin to determine the specificity of bilirubin's pruritic activity (*Figure 1E*). The two metabolites directly epistatic to bilirubin, hemin and biliverdin, did not induce scratching despite also being tetrapyrroles (*Figure 1D*). While hemin, biliverdin, and bilirubin display only minor atomic and electronic differences between them, they vary substantially in their physiochemical properties and structures (*Figure 1E*). To better understand these differences, we performed density functional theory (DFT) calculations (*Becke, 1993*; *Hohenberg and Kohn, 1964*; *Kohn and Sham, 1965*; *Stephens et al., 1994*) to determine the optimal geometry of each metabolite. Unlike in heme and biliverdin, bilirubin's four pyrroles are extended and do not lie in the same plane (*Figure 1E*). DFT calculations revealed that urobilinogen and stercobilin, two bacterial metabolites derived from bilirubin, adopt a similar extended conformation. Both urobilinogen and stercobilin were able to stimulate scratching behavior (*Figure 1D*), indicating that bilirubin's non-polar pyrroles may be important for its pruritic activity.

Patients with jaundice-associated pruritus receive little benefit from antihistamines (*Bergasa, 2014*). Consistent with these clinical findings, the histamine receptor one blocker cetirizine (30 mg/kg, *i.p.*) failed to alleviate scratching behavior in mice injected with bilirubin (*Figure 1—figure supplement 1D*). Furthermore, bilirubin did not elicit a calcium response or induce appreciable histamine release from peritoneal mast cells (*Figure 1—figure supplement 1E–F*).

The Mas-related G-protein coupled receptor (*Mrgpr*) family of receptors is a major mediator of non-histaminergic pruritus (*Han et al., 2013*; *Liu et al., 2012*; *Liu et al., 2009*; *Sikand et al., 2011*). To test whether *Mrgpr*s mediate bilirubin-induced pruritus, we injected mice lacking a cluster of 12 *Mrgpr* genes (Mrgpr-cluster$\Delta^{-/-}$ or Mrgpr-cluster KO) with bilirubin (*Liu et al., 2009*). Mrgpr-cluster KO animals scratched approximately 75% less than wild type (WT) mice, indicating that one or more of the 12 *Mrgpr*s within the cluster mediates bilirubin-induced pruritus (*Figure 1C*).

To identify which *Mrgpr* is sensitive to bilirubin, we individually expressed each of the 12 *Mrgpr*s deleted in the Mrgpr-cluster KO mouse in human embryonic kidney (HEK) 293 cells and monitored changes in intracellular calcium upon applying bilirubin. To ensure we would observe a calcium response following a true ligand-receptor interaction, we expressed the receptors in HEK293 cells stably expressing the G-protein alpha-subunit $G_{\alpha 15}$, a $G_\alpha$ protein that couples GPCRs to intracellular calcium stores via phospholipase C (PLC).

Among the twelve cell lines expressing an *Mrgpr*, only MRGPRA1-expressing cells exhibited a calcium response to bilirubin ($EC_{50}$ of 145.9 μM (*Alemi et al., 2013*)) (*Figure 2A,D*). The same cells that responded to bilirubin also responded to FMRF, an MRGPRA1 agonist (*Dong et al., 2001*). To ensure that bilirubin initiated signaling at MRGPRA1 and not downstream, we pre-treated MRGPRA1-expressing cells with inhibitors of GPCR signaling: the PLC inhibitor U73122 or the $G_{\alpha q}$ inhibitor YM-254890. Both compounds abolished bilirubin-induced calcium responses (*Figure 2B–C*).

In addition to bilirubin, glucuronidated bilirubin is often upregulated in jaundice-associated itch. We assessed whether ditaurate bilirubin, a distinct but similar bilirubin derivative, could activate MRGPRA1. Indeed, ditaurate bilirubin activated MRGPRA1-expressing cells (*Figure 2D*). Hemin failed to activate MRGPRA1 (*Figure 2D*), consistent with our earlier behavioral findings in which hemin did not evoke scratching. No other *Mrgpr* among the 12 that we screened responded to bilirubin (*Figure 2N*, *Figure 2—figure supplement 1*).

The human *MRGPRX* family of receptors has functional similarities between species but have no obvious structural homologs in rodents (*Solinski et al., 2014*; *Zylka et al., 2003*). The mouse *Mrgpra* family is closest in sequence homology to the human *MRGPRX* family (*Dong et al., 2001*; *Lembo et al., 2002*; *Zhang et al., 2005*). Of the four human MRGPRX receptors, only MRGPRX4-expressing cells responded to bilirubin ($EC_{50}$ of 61.9 μM (*Azimi et al., 2017*)) (*Figure 2F,I*). U73122 and YM-254890 inhibited bilirubin-induced calcium responses in MRGPRX4-expressing cells just as with MRGPRA1 (*Figure 2G–H*). Conjugated bilirubin also activated MRGPRX4, whereas hemin had no effect (*Figure 2I*).

To confirm that bilirubin directly binds the identified receptors, we assayed thermophoresis of each receptor in the presence and absence of bilirubin. Thermophoresis of a molecule is affected by physical parameters such as size, charge, and solvation. By extension, the thermophoresis of one molecule is altered when it interacts with another, and can therefore be used to measure interactions between molecules (*Duhr and Braun, 2006*). Using this approach, we determined that bilirubin bound MRGPRA1 with a $K_D$ of 92.9 ± 15 μM and MRGPRX4 with a $K_D$ of 54.4 ± 13 μM (*Figure 2E,J*). Bilirubin exhibited little to no affinity for the closely related BAM8-22 receptor MRGPRC11 (*Figure 2O*). Hemin, which did not activate MRGPRA1 or MRGPRX4 by calcium imaging (*Figure 2D, I*), also did not bind MRGPRA1 or MRGPRX4 (*Figure 2E,J*). Conjugated bilirubin bound both MRGPRA1 and MRGPRX4, although with a lower affinity than unconjugated bilirubin (*Figure 2E,J*). To make certain that bilirubin activates MRGPRA1 and MRGPRX4 upon binding, we measured exchange of guanosine diphosphate (GDP) for guanosine triphosphate (GTP), one of the first events in GPCR signaling. Bilirubin increased GTP binding to MRGPRA1 and MRGPRX4 membrane complexes, but not to MRGPRC11 (*Figure 2K*). To confirm that bilirubin activates MRGPRA1 *in vivo* to trigger itch, we generated an *Mrgpra1* (A1 KO) knockout mouse line using CRISPR-Cas9 (*Jinek et al., 2012*) (*Figure 2—figure supplement 2*). A1 KO animals scratched significantly less than WT mice after exposure to either bilirubin or the established agonist FMRF, demonstrating that Mrgpra1 is functional in adult mice (*Figure 2L–M*). The $K_D$ of bilirubin towards MRGPRA1 and MRGPRX4 suggests that bilirubin likely does not interact with these receptors in healthy individuals.

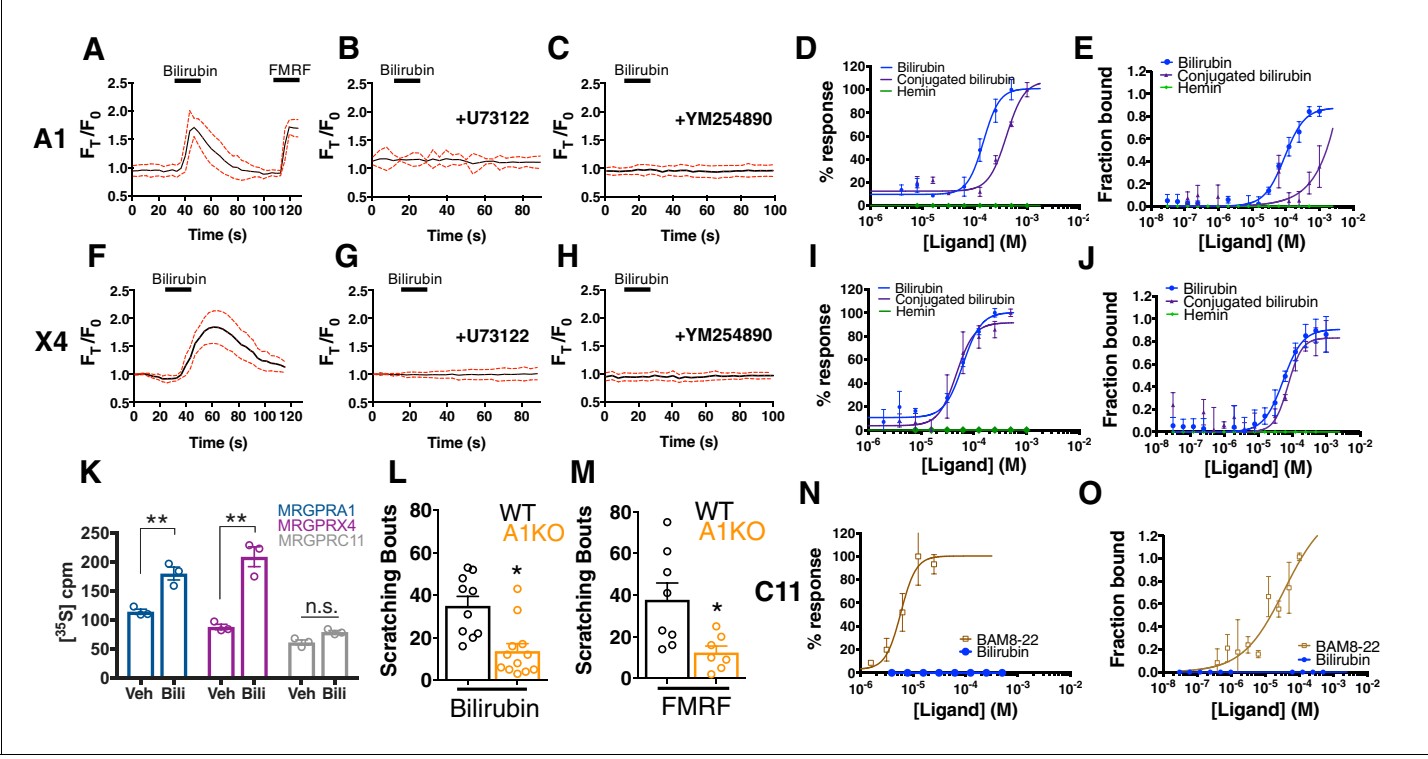

**Figure 2.** Bilirubin activates murine MRGPRA1 and human MRGPRX4. $Ca^{2+}$ imaging and transformed binding isotherms of HEK293 cells stably expressing MRGPRA1 (A-E) or MRGPRX4 (F-J). (A-C, F-H) 50 μM bilirubin was added where indicated by black bars. After 15 s, a 1 min wash was applied. Mean ±95% confidence interval (CI) depicted. n = 10. In (A) 30 μM FMRF was added after washing as indicated by the black bar. In (B-C) and (G-H), cells were pre-incubated with either 10 μM of the PLC inhibitor U73122 or 10 μM of the $G_{\alpha q}$ inhibitor YM254890 for 30 min prior to imaging. Concentration–$Ca^{2+}$ response curves of bilirubin, conjugated bilirubin, and hemin towards (D) MRGPRA1, (I) MRGPRX4, and (N) MRGPRC11 and BAM8-22 towards MRGPRC11, an established peptide ligand. Data are a representative experiment of three independent replicates performed in triplicate, depicted as mean ±s.e.m. Transformed binding isotherms for bilirubin, conjugated bilirubin, and hemin to (E) MRGPRA1, (J) MRGPRX4, and (O) MRGPRC11 and BAM8-22 to MRGPRC11. Data are an average of three independent experiments, depicted as mean ±s.e.m. (K) Bilirubin-stimulated G-protein activity of partially-purified MRGPRA1, MRGPRX4, and MRGPRC11 membrane complexes. $[^{35}S]GTP\gamma S$ binding was measured in the presence of 0.5% DMSO or 50 μM bilirubin. Mean ±s.e.m. depicted. **, p < 0.01; two-tailed unpaired Student's t-test. (L) Scratching bouts from injection of 60 μg of bilirubin in WT and A1 KO animals. WT n = 10, A1 KO n = 12. (M) Scratching bouts from injection of 60 μg of FMRF in WT and A1 KO animals. WT n = 8, A1 KO n = 7. (L-M) Mean ±s.e.m. depicted. Open circles represent individual mice. *, p< 0.05 by two-tailed unpaired Student's t-test.

DOI: https://doi.org/10.7554/eLife.44116.006

The following source data and figure supplements are available for figure 2:

**Source data 1.** Source data for *Figure 2* .
DOI: https://doi.org/10.7554/eLife.44116.009

**Figure supplement 1.** Bilirubin does not activate other itch-associated Mrgprs.
DOI: https://doi.org/10.7554/eLife.44116.007

**Figure supplement 2.** CRISPR deletion of MRGPRA1.
DOI: https://doi.org/10.7554/eLife.44116.008

Additional ligands with nanomolar affinities towards MRGPRA1 or MRGPRX4 may exist that modulate the receptors in normal physiology.

We reasoned that if bilirubin triggers itch through MRGPRA1 and MRGPRX4, bilirubin should activate these receptors in sensory itch neurons. Previous studies have demonstrated that both Mrgpra1 and MRGPRX4 are expressed in sensory neurons within the dorsal root ganglia (DRG) (*Dong et al., 2001*; *Flegel et al., 2015*; *Lembo et al., 2002*). Mrgpra1 is expressed in a subset of adult DRG and trigeminal ganglia (TG) sensory neurons that innervate the skin and ramify in lamina I and II of the spinal cord (*Figure 3A–D*). Bilirubin elicited robust action potentials in small-diameter (<30 μm) WT DRG sensory neurons at a proportion consistent with the percentage of sensory neurons that encode itch (5 of 50). Bilirubin failed to elicit action potentials in A1 KO neurons (0 of 60), suggesting

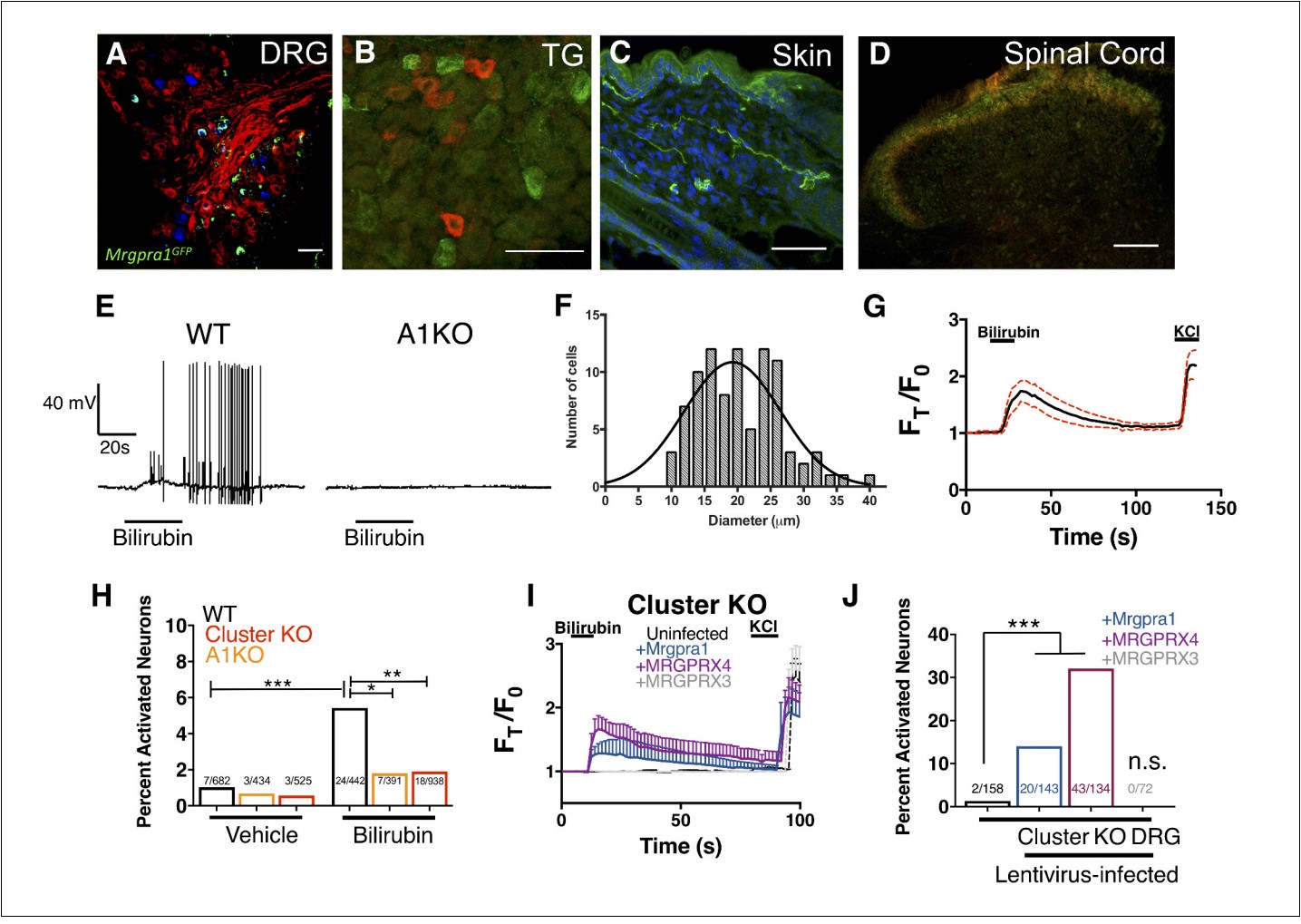

**Figure 3.** Bilirubin activates sensory neurons in an MRGPR-dependent manner. (**A–D**) Confocal microscopy immunofluorescence images of adult mouse tissue sections from Mrgpra1GFP animals with GFP expression under the control of the endogenous *Mrgpra1* locus. (**A**) Mrgpra1 expression in dorsal root ganglia. Green depicts Mrgpra1GFP. Red depicts anti-PLAP antibody staining where PLAP expression is controlled by the endogenous *Mrgprd* locus (*Mrgprd*PLAP). Blue depicts antibody staining against calcitonin gene-related peptide (CGRP). Scale bar is 50 µM. (**B**) Trigeminal ganglia (TG) stained with Mrgpra1GFP (green) and anti-Substance P antibody (red). Scale bar is 50 µm. (**C**) Back skin stained with anti-GFP antibody (green) to visualize Mrgpra1GFP nerve fibers in the dermis. Blue represents DAPI counterstain. Scale bar is 50 µm. (**D**) Spinal cord (SC) (lamina 1 and 2) stained with anti-GFP and IB4-564. Mrgpra1GFP (green) staining was found in lamina two along with IB4 (red) positive terminals. Scale bar is 100 µm. (**E**) Representative whole-cell current-clamp recording of either WT or A1 KO DRG neurons. In WT DRG, bilirubin elicited action potentials in 5 out of 50 small-diameter neurons. In A1 KO DRG, bilirubin elicited action potentials in 0 out of 60 small-diameter neurons. Fisher's exact test p < 0.05. (**F**) Histogram of bilirubin-activated neuronal soma diameter. (**G**) $Ca^{2+}$ imaging of WT DRG neurons. Mean ±95% CI depicted. Compounds applied where indicated by black bars. After a 10 s baseline, 50 µM bilirubin was added. 50 mM KCl was added at the end of each trial. n = 20 neurons. (**H**) Percent activation of WT, A1 KO, and Mrgpr-cluster KO DRG by vehicle and 50 µM bilirubin. *, $P < 0.05$; **, $p < 0.01$; ***, $p < 0.001$; Chi-squared test. A neuron was considered to be activated if ΔF > 0.2 for at least 30 s. (**I**) $Ca^{2+}$ imaging of Mrgpr-cluster KO DRG neurons 48 hr after either mock infection with lentivirus (n = 10) or infected with lentivirus encoding *Mrgpra1* (n = 6), *MRGPRX4* (n = 10), or *MRGPRX3* (n = 20). 50 µM bilirubin was added when indicated by the black bar. After 20 s, a 1 min wash was applied before addition of 50 mM KCl. Compounds applied where indicated by black bars. Mean ±95% CI depicted. n = 10 neurons. (**J**) Percent activation of uninfected, *Mrgpra1*-infected, *MRGPRX4*-infected, and *MRGPRX3*-infected Cluster -/- neurons by bilirubin. ***, p < 0.001. Chi-squared test.

DOI: https://doi.org/10.7554/eLife.44116.010

The following source data is available for figure 3:

**Source data 1.** Source data for *Figure 3*.

DOI: https://doi.org/10.7554/eLife.44116.011

bilirubin activates sensory neurons through MRGPRA1 (*Figure 3E*). Bilirubin-sensitive neurons had an average somal diameter of 20.4 ± 1.3 µm, a diameter characteristic of itch sensory neurons (*Figure 3F*). Applying bilirubin to neurons elicited calcium transients in approximately 5% of WT DRG neurons (*Figure 3G*), whereas significantly fewer sensory neurons from either Mrgpr-cluster KO or A1 KO mice responded (*Figure 3H*). We sought to determine whether expression of either MRGPRA1 or MRGPRX4 was sufficient to render neurons sensitive to bilirubin. To address this question, we infected Mrgpr-cluster KO DRGs with lentivirus carrying either *Mrgpra1*, *MRGPRX4*, or *MRGPRX3*. Bilirubin activated 14% of *Mrgpra1*-and 32% of *MRGPRX4*-transduced Mrgpr-cluster KO DRGs (*Figure 3I–J*). Mrgpr-cluster KO DRGs infected with the control gene *MRGPRX3* did not respond to bilirubin.

Bilirubin-responsive neurons partially overlapped with neurons that responded to 1 mM chloroquine, a ligand for MRGPRA3 that typifies itch sensory neurons (*Han et al., 2013*) (*Figure 4A–C*). To validate that bilirubin activates MRGPRA3-positive itch neurons, we performed calcium imaging on DRG neurons isolated from Tg(*Mrgpra3-Cre);lsl-tdTomato* mice, which express the fluorescent protein tdTomato in *Mrgpra3*-expressing neurons. Bilirubin activated a substantial percentage of *tdTomato*-positive neurons (*Figure 4D*). To confirm that bilirubin activates sensory neurons *in vivo*, we injected 5 µL of vehicle or bilirubin into paws of Tg(*Pirt-Cre);lsl-GCaMP6s* mice, which express the fluorescent calcium reporter GCaMP6s in DRG sensory neurons (*Kim et al., 2014*). Bilirubin, but not vehicle, activated numerous DRG sensory neurons in the paws of GCaMP6s mice (*Figure 4E*). Inhibiting transient receptor potential (TRP) and other $Ca^{2+}$ channels with ruthenium red prevented bilirubin from activating sensory neurons (*Figure 4F–G*) (*Imamachi et al., 2009*; *Liu et al., 2009*; *Roberson et al., 2013*).

We wondered whether chronic elevation of bilirubin *in vivo*, like in cholestasis, stimulates *Mrgpr*-dependent itch. Bile is the primary means by which bilirubin is excreted, and patients with cholestasis exhibit elevated levels of bilirubin and other pruritogenic substances in their blood (*Alemi et al., 2013*). To induce hyperbilirubinemia and model intrahepatic cholestasis, we administered α-napthyl isothiocyanate (ANIT) to mice (*Eliakim et al., 1959*). We treated WT, Mrgpr-cluster KO, and A1 KO animals with 25 mg/kg ANIT for five days before assessing spontaneous itch (*Figure 5A*). WT, Mrgpr-cluster KO, and A1 KO animals exhibited equivalently severe hepatocellular injury, judged by increases in plasma bilirubin, bile acids, alkaline phosphatase (ALP), aspartate aminotransferase (AST), alanine aminotransferase (ALT), and gamma-glutamyl transferase (GGT) (*Figure 5D–E*, *Figure 5—figure supplement 1A–D*).

As expected, ANIT treatment significantly increased pruritus in all animals (*Figure 5B*). However, Mrgpr-cluster KO and A1 KO mice scratched markedly less than WT mice (*Figure 5B*), suggesting that MRGPRA1 mediates a component of hepatobiliary pruritus. In humans, bile acids, endogenous opioids, and LPA are often increased in cholestatic sera and have been shown to mediate pruritus (*Alemi et al., 2013*; *Bergasa et al., 1998*; *Bergasa et al., 1992*; *Kremer et al., 2010*). The serum of ANIT-treated animals exhibited elevated bilirubin and bile acids (*Figure 5D–E*), whereas neither the endogenous opioid peptide met-enkephalin (*Thornton and Losowsky, 1989a*; *Thornton and Losowsky, 1989b*) nor the LPA-producing enzyme autotaxin were elevated (*Figure 5—figure supplement 1E–F*). To assess whether other cholestatic pruritogens act at MRGPRs in mice, we injected WT and Mrgpr-cluster KO with deoxycholic acid (a bile acid), opiates, and LPA. These other cholestatic pruritogens elicited equivalent degrees of scratching in WT and Mrgpr-cluster KO animals (*Figure 5—figure supplement 2A–D*). Mrgprs are promiscuous receptors. It should be noted that there remain multiple bile acids, LPA molecules, and opiates which remain untested that may be agonists of Mrgprs. Based on this data, we hypothesized that Mrgpr-cluster KO and A1 KO mice scratched less with ANIT because MRGPRA1 mediates bilirubin-induced itch.

To determine whether bilirubin is activating MRGPRA1 to stimulate itch in cholestasis, we induced cholestasis in a mouse that lacks the biosynthetic enzyme for bilirubin, biliverdin reductase (BVR KO) (*Kutty and Maines, 1981*) (*Figure 1E*, *Figure 5—figure supplement 3A*). BVR KO mice lack *Blvra*, the gene that encodes for bilirubin reductase. BVR KO mice do not have detectable levels of bilirubin in plasma (*Figure 5—figure supplement 3B–D*). When treated with ANIT, BVR KO mice scratched significantly less than WT mice (*Figure 5C*). Plasma levels of bile acids, ALP, AST, ALT, GGT, met-enkephaline, and autotaxin were indistinguishable between treated BVR KO animals and WT controls (*Figure 5—figure supplement 1A–D*). Their diminished response to ANIT is not due to

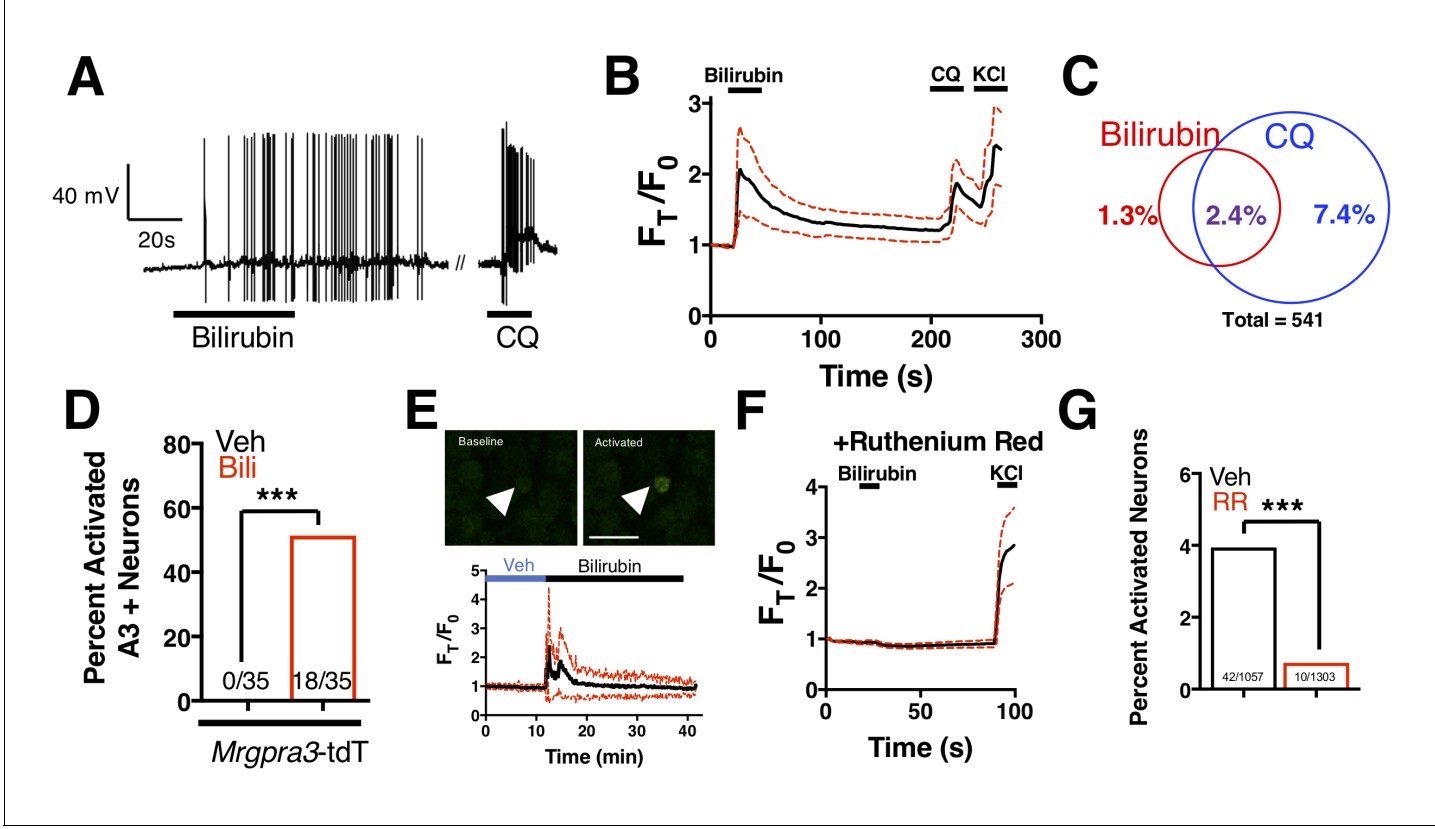

**Figure 4.** Bilirubin activates a population of small-diameter, chloroquine-sensitive sensory neurons in a TRP channel dependent mechanism. (A) A representative whole-cell current-clamp recording of a WT DRG neuron responsive to addition of both 50 µM bilirubin and 1 mM chloroquine (CQ). (B) $Ca^{2+}$ imaging of WT DRG neurons. Mean ±95% CI depicted. Compounds applied where indicated by black bars. After a 10 s baseline, 50 µM bilirubin was added. After 20 s, a 3 min wash was applied before 1 mM chloroquine was added. After 15 s, 50 mM KCl was added. n = 10 neurons. (C) Venn diagram of total neurons activated by either bilirubin and/or chloroquine (Bilirubin alone = 7, Chloroquine = 40, Overlap = 13). (D) Percent activation of Tg(*Mrgpra3-Cre*);lsl-tdTomato neurons as assessed by calcium imaging with vehicle, 1 mM Chloroquine, or 50 µM bilirubin. (E) *In vivo* $Ca^{2+}$ imaging of Pirt-Cre; lsl-GCaMP6s animals. Briefly, a surgery was performed to expose L4 DRG. Baseline measurements were taken before a vehicle injection in ipsilateral paw and subsequent injection of 5 µL of 100 µM bilirubin. Depicted is a representative $Ca^{2+}$ imaging trace of bilirubin-activated neurons, n = 20, identified by post hoc imaging analysis. The black trace is the mean $F_t/F_0$ and red dotted lines represent 95% confidence intervals. (F) Neurons were incubated with 10 µM ruthenium red for 10 min before application of 50 µM bilirubin. (G) Percent activation of WT neurons, incubated with either vehicle or 10 µM ruthenium red, by 50 µM bilirubin.

DOI: https://doi.org/10.7554/eLife.44116.012

The following source data is available for figure 4:

**Source data 1.** Source data for *Figure 4*.
DOI: https://doi.org/10.7554/eLife.44116.013

aberrant itch circuits, as BVR KO mice scratched normally when injected with either chloroquine or exogenous bilirubin (*Figure 5—figure supplement 3E–F*).

To confirm that the observed differences in cholestatic pruritus were not just specific to ANIT, we administered the hepatotoxin cyclosporin A to WT, A1 KO, and BVR KO mice (*Laupacis et al., 1981*). We treated mice with either 50 mg/kg cyclosporin A or vehicle for eight days before assessing spontaneous itch (*Figure 5F*). Cyclosporin A induced spontaneous itch in WT animals, whereas A1 KO and BVR KO mice again scratched significantly less than WT mice (*Figure 5G–H*).

Notably, we found that plasma bilirubin correlates poorly with cholestatic itch in patients and in cholestatic animals (*Figure 5I*). We hypothesized that the levels of bilirubin in the skin would correlate better with itch than serum bilirubin largely because bilirubin likely binds and activates the sensory neurons in the skin. Unlike with serum, skin bilirubin appears to be a much stronger predictor of itch severity in mice (*Figure 5J*). This is consistent with the anatomical distribution of itch sensory neurons and may explain why studies aimed at identifying plasma pruritogens that correlate with

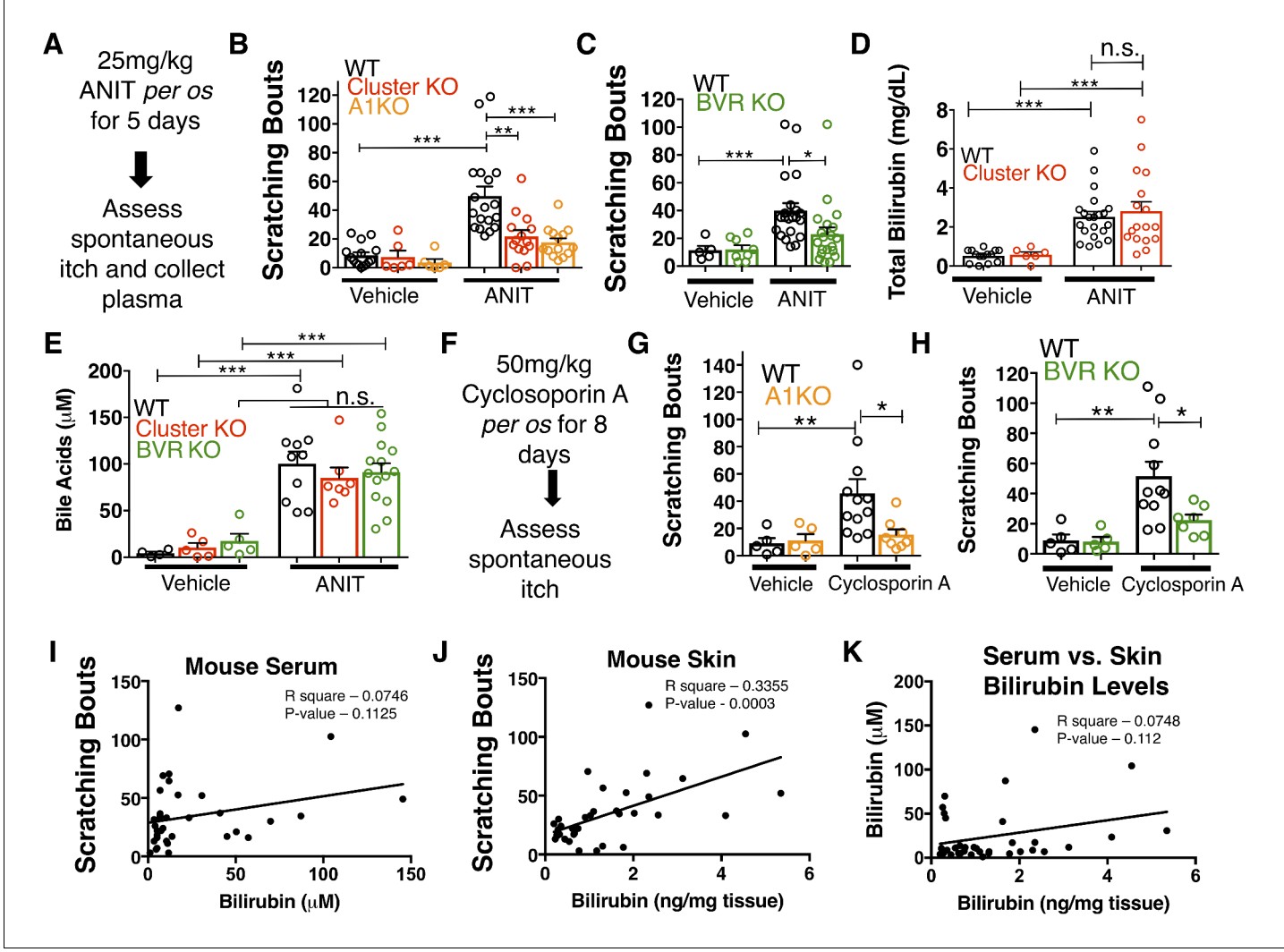

**Figure 5.** MrgprA1 KO, Mrgpr-cluster KO⁻, and BVR KO animals exhibit decreased cholestatic pruritus. (**A**) Experimental flowchart for ANIT model of cholestasis. (**B**) Scratching bouts for vehicle and ANIT-treated mice among WT, Mrgpr-cluster KO, and A1 KO groups. Bouts were assessed in a 30 min period. For the vehicle cohort: WT n = 15, Mrgpr-cluster KO n = 6, A1 KO n = 6. For ANIT cohort: WT n = 20, Mrgpr-cluster KO n = 14, A1 KO n = 14. (**C**) Scratching bouts for vehicle and ANIT-treated animals among WT and BVR KO groups. Bouts were assessed in a 30 min period. For the vehicle cohort: WT n = 5 and BVR KO n = 8. For ANIT cohort: WT n = 21 and BVR KO n = 20. (**D**) Plasma bilirubin levels (mg/dL) from WT and Mrgpr-cluster KO ANIT-treated and vehicle-treated animals. For the vehicle cohort: WT n = 14, Mrgpr-cluster KO n = 6. For the ANIT cohort: WT n = 21, Mrgpr-cluster KO n = 17. (**E**) Plasma bile acid levels (μM) from ANIT-treated and vehicle-treated animals. For the vehicle cohort: WT n = 4, Mrgpr-cluster KO n = 5, BVR KO n = 5. For the ANIT cohort: WT n = 10, Mrgpr-cluster KO n = 7, BVR KO n = 14. (**F**) Experimental flowchart for Cyclosporin A model of cholestasis. (**G**) Scratching bouts for vehicle and Cyclosporin A-treated WT and A1 KO animals. For the vehicle cohort: n = 5 for all. For Cyclosporin A cohort: WT n = 10 and A1 KO n = 8. (**H**) Scratching bouts from vehicle and Cyclosporin A treated WT and BVR KO animals. For the vehicle cohort: n = 5. For Cyclosporin A cohort: WT n = 11 and BVR KO n = 7. (**I**) Correlation of serum bilirubin levels from cholestatic animals and scratching bouts. Line of best fit: Y = 0.23 (X) + 28.78. (**J**) Correlation of skin bilirubin levels from cholestatic animals and scratching bouts. Line of best fit: Y = 12.34 (X) + 16.7. (**K**) Correlation of skin and serum bilirubin levels from cholestatic animals. Line of best fit: Y = 7.015(X) – 14.52. (**A–H**) Mean ±s.e.m. depicted. Open circles represent individual data points. *, p < 0.05; **, p < 0.01; ***, p < 0.001 by unpaired two-tailed Student's t-test.

DOI: https://doi.org/10.7554/eLife.44116.014

The following source data and figure supplements are available for figure 5:

**Source data 1.** Source data for *Figure 5*.
DOI: https://doi.org/10.7554/eLife.44116.018

**Figure supplement 1.** Plasma levels of pathological markers of liver injury are similar between WT, Mrgpr-clusterΔ⁻/⁻, A1 KO, and BVR KO animals.
DOI: https://doi.org/10.7554/eLife.44116.015

**Figure supplement 2.** Mrgpr-cluster KO animals have intact itch to other cholestatic pruritogens and bilirubin synergism with chloroquine itch.
DOI: https://doi.org/10.7554/eLife.44116.016

*Figure 5 continued*

**Figure supplement 3.** BVR KO and A1 KO animals have intact itch circuits.

DOI: https://doi.org/10.7554/eLife.44116.017

itch severity may have missed bilirubin. Secondarily, we find that plasma bilirubin does not correlate well with skin bilirubin, further suggesting that plasma bilirubin may be a poor predictor of itch severity and may not necessarily serve as a proxy for skin bilirubin (*Figure 5K*). The amount of bilirubin in the skin is likely affected by several factors and equilibria, such as serum albumin.

We assessed whether pharmacologically antagonizing MRGPRs could alleviate cholestatic itch. Recently, a 3-amino acid peptide, QWF, was identified as an MRGPRA1 antagonist (*Azimi et al., 2016*). QWF abolished bilirubin-associated calcium signaling in MRGPRA1-expressing cells with an IC$_{50}$ of 2.9 μM [1, 5] (*Figure 6A*). Mirroring its pharmacology *in vitro*, co-injecting 0.25 mg/kg QWF with bilirubin significantly alleviated pruritus associated with bilirubin (*Figure 6B*). QWF specifically antagonized bilirubin, as it did not attenuate chloroquine-MRGPRA3 associated itch (*Figure 6C*).

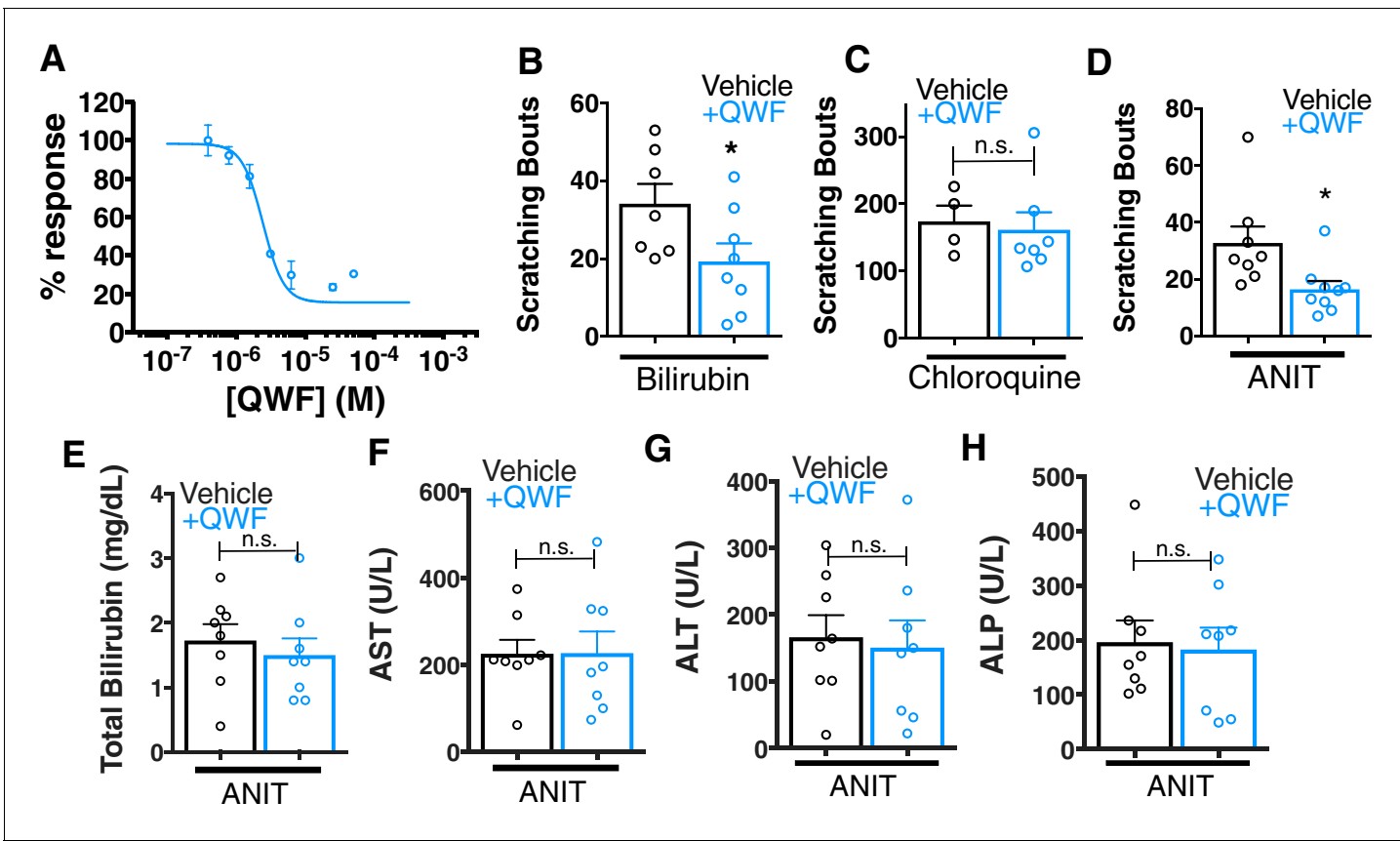

**Figure 6.** QWF treatment blocks bilirubin activation of Mrgpra1 and cholestatic pruritus. (**A**) Concentration-response curve for bilirubin induced Ca$^{2+}$ signal in MRGPRA1-expressing HEK cells. 200 μM bilirubin was maintained in competition with indicated doses of QWF. Mean ±s.e.m. depicted. n = 3 replicates in duplicate. (**B–C**) Scratching bouts from (**B**) 60 μg (100 mL of 1 mM) bilirubin or (**C**) 150 μg chloroquine co-injected with either vehicle or 1 mg/kg QWF. After injection, the number of scratching bouts in 30 min was assessed. For bilirubin: Vehicle n = 7, QWF n = 8. For chloroquine: Vehicle n = 4, QWF n = 7. (**D**) Scratching bouts from WT ANIT-treated animals. Either vehicle or 1 mg/kg QWF was delivered *i.p.* Vehicle n = 8, QWF n = 9. (**E–H**) Plasma (**E**) bilirubin, (**F**) AST, (**G**) ALT, and (**H**) ALP levels from of vehicle and QWF-dosed WT animals that have undergone ANIT liver injury. (**B–H**) Mean ±s.e.m. depicted. Open circles represent independent data points. n.s., not significant; *, p < 0.05 by two-tailed unpaired Student's *t*-test.

DOI: https://doi.org/10.7554/eLife.44116.019

The following source data is available for figure 6:

**Source data 1.** Source data for *Figure 6*.

DOI: https://doi.org/10.7554/eLife.44116.020

Lastly, we evaluated whether the MRGPRA1 antagonist QWF could alleviate cholestatic pruritus. We dosed WT animals with ANIT as previously described, but intraperitoneally injected mice with either vehicle or 1 mg/kg QWF thirty minutes prior to behavioral analysis. Mice treated with QWF scratched significantly less than vehicle-treated animals (*Figure 6D*). QWF treatment did not change plasma levels of total bilirubin, AST, ALT, or ALP, suggesting that QWF treatment did not alter the underlying liver pathology (*Figure 6E–H*).

Nasobiliary drainage is the most effective treatment for cholestatic pruritus (*Hegade et al., 2016*). Based on this clinical observation, we predicted that plasma isolated from cholestatic animals would elicit pruritus (*Figure 7A*). Indeed, plasma from WT animals with cholestasis elicited itch when injected into naïve WT animals (*Figure 7B*). Cholestatic plasma isolated from BVR KO mice, which lacks bilirubin (*Figure 5—figure supplement 3B–D*), elicited significantly fewer scratches than WT cholestatic plasma (*Figure 7B*). The levels of ALP, AST, and ALT were indistinguishable between WT and BVR KO cholestatic plasma (*Figure 5—figure supplement 1A–D*), presumably because ANIT

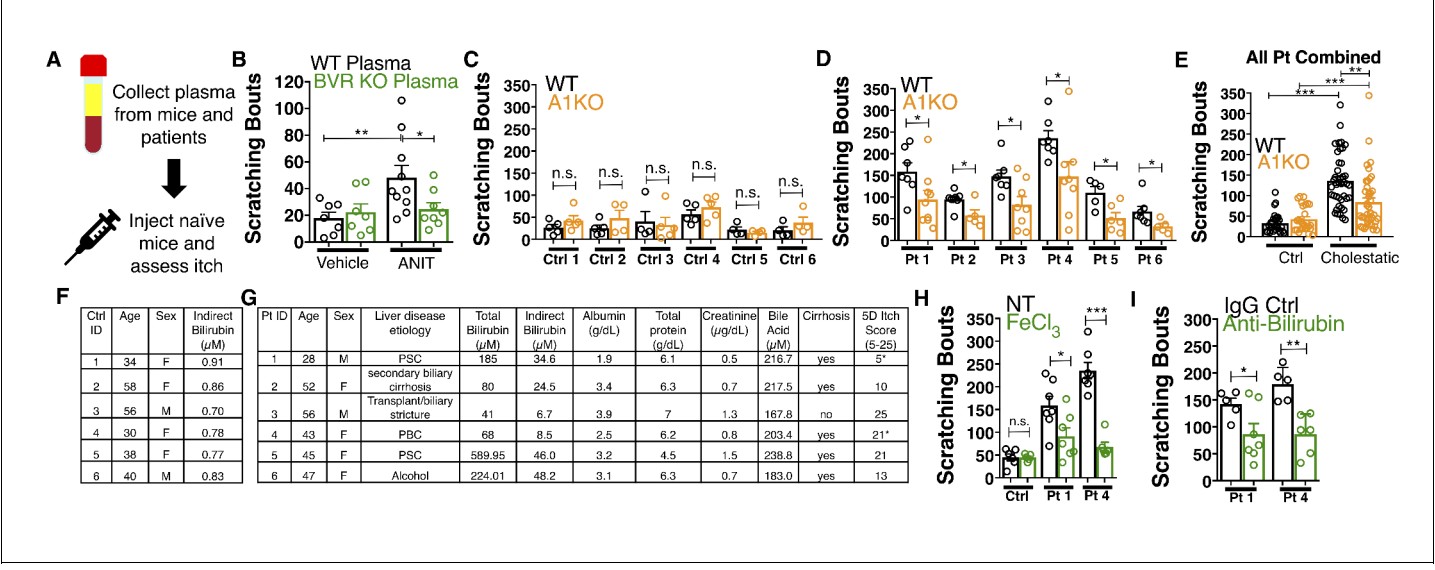

**Figure 7.** Bilirubin from mouse and human cholestatic plasma contributes to pruritus in a Mrgpra1-dependent manner. (A) Experimental flowchart of plasma injection assay. (B) Scratching bouts from WT mice injected with either vehicle- or ANIT-treated plasma from WT and BVR KO animals. For the vehicle plasma cohort: n = 7. For cholestatic ANIT-treated plasma: WT n = 10 and BVR KO n = 8. (C) Scratching bouts from either WT or A1 KO mice injected with control donor plasma. Ctrl 1, WT n = 5, A1 KO n = 5. Ctrl 2, WT n = 5, A1 KO n = 5. Ctrl 3, WT n = 5, A1 KO n = 4. Ctrl 4, WT n = 4, A1 KO n = 5. Ctrl 5, WT n = 4, A1 KO n = 4. Ctrl 6, WT n = 4, A1 KO n = 4. (D) Scratching bouts from either WT or A1 KO mice injected with either cholestatic patient plasma. Patient 1, WT n = 7, A1 KO n = 9. Patient 2, WT n = 8, A1 KO n = 5. Patient 3, WT n = 7, A1 KO n = 8. Patient 4, WT n = 6, A1 KO n = 8. Patient 5, WT n = 5, A1 KO n = 6. Patient 6, WT n = 7, A1 KO n = 5. (E) Scratching bouts from either WT or A1 KO mice injected with plasma collected from all tested control donor samples and all tested cholestatic patient samples. For Control: WT n = 27, A1 KO n = 27. For cholestatic patient: WT n = 41, A1 KO n = 41. (F–G) Biochemical characteristics of patient and control plasma. (F) Age, sex, and bilirubin levels of control plasma collected. All control plasma donors did not suffer from any chronic itch condition. (G) 5D itch questionnaire was administered at time of plasma collection. Asterisk denotes patients taking anti-pruritic medication at time of plasma collection and questionnaire administration. Patient one was taking sertraline (100 mg QD) and Patient four was taking Gabapentin (800 mg TID). A score of 25 represents the maximum level of itchiness. (H) Scratching bouts from mice injected with either untreated (NT) control human plasma, FeCl$_3$-treated control human plasma, NT cholestatic patient 1 and 4 plasma (same data from Patient WT data in (D), or FeCl$_3$-treated patient plasma. For control plasma, NT n = 6 and FeCl$_3$n = 5. For Patient one plasma, NT n = 7 and FeCl$_3$n = 7. For Patient four plasma, NT n = 7 and FeCl$_3$n = 6. (I) Scratching bouts from mice injected with either normal rabbit IgG –treated patient plasma or anti-bilirubin IgG – treated patient plasma. For Patient 1, Normal IgG n = 5, Anti-bilirubin n = 7. For Patient 4, Normal IgG n = 5, Anti-bilirubin n = 6. (B–I) Mean ±s.e.m. depicted. Open circles represent independent data points. n.s., not significant; *, p<0.05; **, p<0.01; ***, p<0.001 by unpaired two-tailed Student's t-test.

DOI: https://doi.org/10.7554/eLife.44116.021

The following source data and figure supplement are available for figure 7:

**Source data 1.** Source data for *Figure 7*.
DOI: https://doi.org/10.7554/eLife.44116.023

**Figure supplement 1.** FeCl$_3$ and anti-bilirubin antibody depletion of plasma bilirubin.
DOI: https://doi.org/10.7554/eLife.44116.022

induced similar hepatotoxicity in WT and BVR KO mice. Instead, BVR KO plasma likely results in less pruritus because it lacks bilirubin.

We also isolated plasma from six patients suffering from various conditions that result in hyperbilirubinemia and six age- and sex-matched control patients (*Figure 7C–G*). All six cholestatic patients' plasma evoked itch in WT animals (*Figure 7D*). When injected into A1 KO animals, each patient's plasma elicited less itch (*Figure 7D*). Compared to plasma from itchy patients, plasma from healthy donors with low levels of bilirubin elicited less itch in WT animals (*Figure 7C*). Plasma from healthy donors injected into A1 KO animals elicited similar scratching behavior (*Figure 7C*). To assess whether removing bilirubin from cholestatic plasma may be therapeutic, we depleted bilirubin both by selective oxidation with $FeCl_3$ or an anti-bilirubin antibody and again evaluated its pruritic capacity. We verified depletion of bilirubin by HPLC (*Figure 7—figure supplement 1A–B*). Injecting WT mice with plasma (cholestatic patients 1 and 4) treated with $FeCl_3$ or an anti-bilirubin antibody evoked less pruritus compared to untreated or control IgG-treated patient plasma (*Figure 7H–I*).

## Discussion

To date, bilirubin has largely been considered a neonatal neurotoxin or an inert biomarker in disease. Our results reveal that bilirubin itself is a pruritogen that evokes itch by binding and activating MRGPRs on sensory neurons and may be an overlooked source of some patients' unrelenting itch. The $K_D$ of bilirubin towards MRGPRA1 and MRGPRX4 suggests that bilirubin likely only interacts and activates these receptors in individuals with markedly elevated bilirubin and not in healthy people. More narrowly in hepatobiliary diseases such as cholestasis, our data supports a model in which bilirubin is one of several pruritogens that contributes to itch. Specifically, we find that genetically removing either MRPGRA1 (A1 KO) or bilirubin (BVR KO) both strongly attenuated itch. However, these mutant mice still exhibit greater itch compared to untreated mice. This suggests that other pruritogens contribute to itch alongside bilirubin in hepatobiliary disease, consistent with the diverse and complex presentations of patients suffering from conditions such as chronic pruritus. Other responsible pruritogens could include bile acids, endogenous opioids, and LPA. While depleting bilirubin in jaundiced patients like in mice may be effective in reducing itch, not every patient who suffers from hepatobiliary pruritus is jaundiced. Accordingly, identifying and depleting other pruritogens may similarly reduce itch.

Although our findings directly illustrate that bilirubin is pruritic, it is also clear that not every patient with jaundice experiences itch. For example, patients with genetic hyperbilirubinemias such as Dubin-Johnson syndrome, a disorder involving mutations in the bilirubin transporter ABCC2, or Crigler-Najjar Type 1, a disorder involving mutations in the bilirubin glucuronosyltransferase UGT1A1, rarely complain of pruritus (*Levitt and Levitt, 2014*; *van der Veere et al., 1996*). Moreover, neonates can have high levels of bilirubin in their skin but not itch. Bilirubin thus appears to exert selective pruritic activity in certain contexts, which we hypothesize may derive from its dynamic biophysical behavior and complex network of interactions.

In isolated genetic hyperbilirubinemias, few – if any – other organic metabolites are elevated. In contrast, several other metabolites are elevated in addition to bilirubin in cholestasis, many of which may alter the equilibrium between bilirubin and albumin (*Alemi et al., 2013*; *Jacobsen and Brodersen, 1983*; *Kalir et al., 1990*; *Kozaki et al., 1998*). Moreover, bilirubin's affinity for albumin and other lipoproteins is disrupted by numerous agents in bile that are upregulated specifically in cholestasis. Bilirubin also exhibits distinct chemical behavior in cholestatic serum, and several groups have suggested that bile acids affect bilirubin's solubility and conformation (*Ostrow and Celic, 1984*; *Rege et al., 1988*). Accordingly, it is reasonable to speculate that bilirubin is more likely bound to albumin in isolated hyperbilirubinemias than in cholestasis, and is therefore less likely to enter the skin in conditions like Dubin-Johnson. Notably, a predictive metrics for cholestatic pruritus is the Mayo risk score, which considers both serum bilirubin and albumin levels (*Talwalkar et al., 2003*). The Mayo risk score predicts that itch burden increases with increasing bilirubin and decreasing albumin levels; in such circumstances, bilirubin is less likely bound to albumin and is free to enter other tissues such as the skin. Crigler-Najjar patients may be even less likely to complain of itch than Dubin-Johnson patients because the standard treatments for Crigler-Najjar (phenobarbital and light therapy) may directly interfere with bilirubin's pruritic activity. Specifically, light therapy induces photoisomerization and/or photolysis of bilirubin, which alter its structure and activity. Phenobarbital

itself acts by broadly decreasing neural excitability, and may dampen itch circuitry alongside other central nervous system circuits. Notwithstanding these questions, our results suggest that blocking MRGPRX4 may offer relief to those suffering from jaundice and/or cholestatic-associated pruritus.

# Materials and methods

## Key resources table

| Reagent type (species) or resource | Designation | Source or reference | Identifiers | Additional information |
|---|---|---|---|---|
| Gene (M. Musculus) | Mrgpra1 | N/A | MGI:3033095 | |
| Gene (H. Sapiens) | MRGPRX4 | N/A | Ensembl: ENSG00000179817 | |
| Genetic reagent (M. Musculus) | Mrgpr-cluster KO | PMID: 20004959 | MGI:2684085 | |
| Genetic reagent (M. Musculus) | Mrgpra1GFP | PMID: 11551509 | MGI:3033095 | |
| Genetic reagent (M. Musculus) | MrgprdPLAP | PMID: 17618277 | MGI:3033142 | |
| Genetic reagent (M. Musculus) | Tg(Mrgpra3-Cre) | PMID: 23263443 | MGI:5506535 | |
| Genetic reagent (M. Musculus) | Tg(Pirt-Cre) | PMID: 24462040 | MGI:2443635 | |
| Genetic reagent (M. Musculus) | Rosa26-lslGCaMP6s | Jackson Labs | Stock No: 024106 | |
| Genetic reagent (M. Musculus) | Rosa26-lslTdTomato | Jackson Labs | Stock No: 007914 | |
| Genetic reagent (M. Musculus) | Mrgpra1 KO | Newly generated | MGI:3033095 | Generated by CRISPR Cas9 by Johns Hopkins Transgenics Core on the C57Bl/6J background |
| Genetic reagent (M. Musculus) | BVR KO | Newly generated | MGI:88170 | Generated by Ozgene on the C57Bl/6J background |
| Chemical compound, drug | bilirubin IXα | Frontier Scientific | Catalog No. B584-9 | |
| Chemical compound, drug | a-naphthyl isothiocyanate | Sigma | Catalog No. N4525 | |
| Chemical compound, drug | Biliverdin | Sigma | Catalog No. 30891 | |
| Chemical compound, drug | Chloroquine | Sigma | Catalog No. C6628 | |
| Chemical compound, drug | Compound 48/80 | Sigma | Catalog No. C2313 | |

*Continued on next page*

*Continued*

| Reagent type (species) or resource | Designation | Source or reference | Identifiers | Additional information |
|---|---|---|---|---|
| Chemical compound, drug | Cyclosporin A | Sigma | Catalog No. 30024 | |
| Chemical compound, drug | Hemin | Sigma | Catalog No. H9039 | |
| Chemical compound, drug | U73122 | Selleck Chemicals | Catalog No. S8011 | |
| Chemical compound, drug | YM254890 | Wako Chemicals | Catalog No. 253–00633 | |
| Chemical compound, drug | Ruthenium Red | Sigma | Catalog No. R2751 | |
| Chemical compound, drug | Fluo-4 AM | Molecular Probes | Catalog No. 20550 | |
| Chemical compound, drug | Fura-2 AM | Thermo Fisher | Catalog No. F1201 | |
| Chemical compound, drug | BOC-GLN-D-(FORMYL)TRP-PHE-BENZYLESTER | Sigma | Catalog No. B5431 | |
| Chemical compound, drug | Bilirubin ditaurate | Lee Biosciences | Catalog No. 910–12 | |
| Chemical compound, drug | Cetirizine | Tocris Biosciences | Catalog No. 2577 | |
| Chemical compound, drug | Stercobilin | Santa Cruz Biotechnology | Catalog No. sc-264326 | |
| Chemical compound, drug | Urobilinogen | Santa Cruz Biotechnology | Catalog No. sc-296690 | |
| Chemical compound, drug | GS-IB4 - 568 | Invitrogen | I-21412 | (1:500) |
| Peptide, recombinant protein | Human Serum Albumin | Sigma | Catalog No. A9511 | |
| Peptide, recombinant protein | BAM8-22 | Sigma | Catalog No. SML0729 | |
| Peptide, recombinant protein | FMRF peptide | Sigma | Catalog No. N3637 | |
| Peptide, recombinant protein | Fibronectin | Sigma | Catalog No. F0895 | |
| Antibody | anti-GFP (rabbit polyclonal) | Molecular Probes | A-11122; RRID:AB_221569 | (1:1000) |

*Continued on next page*

*Continued*

| Reagent type (species) or resource | Designation | Source or reference | Identifiers | Additional information |
|---|---|---|---|---|
| Antibody | anti-CGRP (rabbit polyclonal) | Peninsula | T-4239; RRID:AB_518150 | (1:1000) |
| Antibody | anti-Substance P (rat monoclonal) | Abcam | M09205; RRID:AB_305866 | (1:250) |
| Antibody | Goat anti-rabbit (Alexa Fluor 568) | Molecular Probes | A11011; RRID:AB_143157 | (1:1000) |
| Antibody | Goat anti-rabbit (Alexa Fluor 488) | Molecular Probes | A11008; RRID:AB_143165 | (1:1000) |
| Antibody | Goat anti-Rat (Alexa Fluor 647) | Invitrogen | A-21247; RRID:AB_141778 | (1:500) |

## Animal care and use

All experiments were performed in accordance with protocols approved by the Animal Care and Use Committee at the Johns Hopkins University School of Medicine.

## Isolation of human plasma

Plasma from patients suffering from hyperbilirubinemia, specifically cholestasis, was isolated under a protocol approved by the Institutional Review Board at the Johns Hopkins University School of Medicine (Study number: IRB00154650). Control plasma was isolated from donors who did not exhibit kidney or liver disease, had no complaints of itch, and were free from any detectable viral infection (HCV, HBV, HIV). Both cholestatic and control plasma were isolated under protocols approved by the Institutional Review Board at the Johns Hopkins University School of Medicine (Cholestasis Study number: IRB00154650; Control study number: NA_00013177, the Johns Hopkins Department of Dermatology Patient Database and Tissue Bank). Whole blood was collected into PAXgene tubes (PreAnalytiX 761115) and centrifuged for 5 min at 300 g. Plasma was then collected, aliquoted, and stored at −20°C until experimentation. At time of plasma collection, a 5D itch questionnaire was administered.

## Molecules and preparation

The following molecules were used: bilirubin IXα (Frontier Scientific). α-naphthyl isothiocyanate (ANIT, Sigma), biliverdin (Sigma), chloroquine (Sigma), compound 48/80 (Sigma), cyclosporin A (Sigma), hemin (Sigma), human serum albumin (HSA, Sigma), BAM8-22 (Sigma), BOC-GLN-D-(FORMYL)TRP-PHE-BENZYLESTER (QWF, Sigma), bilirubin ditaurate (Lee Biosciences). cetirizine (Tocris Biosciences), stercobilin (Santa Cruz Biotechnology), urobilinogen (Santa Cruz Biotechnology), FMRF peptide (Sigma), cholera toxin (Santa Cruz Biotechnology), U73122 (Santa Cruz Biotechnology), YM-254890 (Wako Chemicals), pertussis toxin (Fisher Scientific), fibronectin (Sigma), ruthenium red (Sigma), Fluo 4-AM (Molecular Probes), and Fura 2-AM (Molecular Probes).

Bilirubin is highly susceptible to oxidation and photolysis. Accordingly, bilirubin was freshly prepared just prior to each experiment in either DMSO or 0.1 M NaOH and then maintained in the dark. For calcium imaging analyses, bilirubin was diluted into calcium imaging buffer a few seconds before use. Final concentration of DMSO in all applicable tested solutions was <0.5%. ANIT and cyclosporin A were dissolved in olive oil and prepared freshly as needed. Urobilinogen and stercobilin were dissolved in phosphate buffered saline and adjusted to a pH of 7.4 before being stored at −20°C in 100 µl aliquots until needed. To maintain the integrity of bilirubin in human plasma samples, samples were stored at −80°C until use. Plasma stocks were maintained in the dark to minimize photolysis during injection or experimental manipulation. Plasma bilirubin was evaluated by HPLC as described above. All other drugs were prepared as 100 µl – 1,000 µl aliquots and stored at −20°C before thawing at 4°C. Freeze/thaw cycles were avoided whenever possible. To remove

microprecipitates, we centrifuged our bilirubin solutions at 21,000 g for 20 min to ensure that bilirubin was in solution. Whenever physiologically and experimentally reasonable, we excluded divalent cations in our in vitro biophysical experiments.

## Behavioral studies

All applicable behavioral tests were performed and analysed with the experimenter blind to genotype. All mice used were 8–12 week old males and females (20 to 30 g) that had either been generated on a C57BL/6J background or backcrossed to C57BL/6J mice for at least 10 generations. There were no significant differences in itch between male and female mice. All itch behavior experiments were performed between 8 a.m. and 12 p.m. On the day before the experiment, animals were placed in the test chamber for 30 min before being subjected to a series of three mock injections with 5 min break periods in between. On the day of the experiment, animals were first allowed to acclimatize to the test chamber for 10 min before injection. Pruritic compounds were subcutaneously injected into the nape of the neck (50 µL volume) or cheek (10 µL volume), and scratching behavior was observed for 30 min. A bout of scratching was defined as a continuous scratching movement with either hindpaw directed at the area of the injection site. In the cheek injection model, a wipe was defined as a single forepaw stroking the site of the injection. Use of both forepaws on the face or cheek was considered as grooming behavior. Scratching behavior was quantified by counting the number of scratching bouts at 5 min intervals over the 30 min observation period. Wiping was quantified at 2 min intervals over a ten-minute observation period. For H1R block, 30 mg/kg of cetirizine HCl (pH 7.4) was given intraperitoneally thirty minutes prior to injection of bilirubin. Licking behavior was quantified in seconds and identified as the licking of the toes or footpad of the hind paw site of injection that was neither preceded nor followed by licking of any other portion of the body. For QWF co-injection experiments, either 100 µM (for bilirubin) or 500 µM (for chloroquine) QWF was injected in the same volume as solubilized pruritogen.

## Generation of knock-in and knock-out mice

Mrgpr-clusterΔ$^{-/-}$ mice, Mrgpra1$^{GFP}$ mice, and Mrgprd$^{PLAP}$ were generated as previously described (Dong et al., 2001; Liu et al., 2009; Liu et al., 2007). Tg(Mrgpra3-Cre) mice were generated as previously described (Han et al., 2013). Tg(Pirt-Cre) mice were generated as previously described via homologous recombination (Kim et al., 2014). Rosa26-LoxP-STOP-LoxP (lsl)-GCaMP6s mice were purchased from Jackson Labs.

Lsl-tdTomato mice (Ai9, 007909) were purchased from Jackson Labs. Mrgpra1$^{-/-}$ mice were generated using CRISPR-Cas9 on the C57BL/6 background using the following guide RNA sequence: TTCCCAGCAGCACCTGTGCAGGG. Blvra$^{-/-}$ mice were generated at Ozgene (Australia) on a C57BL/6J background.

## Quantum mechanical calculations

DFT calculations were performed with Spartan 16 and modelled with wxMacMolPlt. Geometry optimizations and single point energy calculations were carried out with DFT-Hartree Fock hybrid B3LYP theory with the 6-31G(d) basis set. Energies were calculated at ground state in the gas phase 298 K.

## Calcium imaging and analysis

All cultured cells were maintained in DMEM supplemented with 10% FBS and 1% Penicillin/Streptomycin at 37°C, 5% $CO_2$. All data included were generated from cells tested for mycoplasma and found to be negative. Cells were imaged in calcium imaging buffer (CIB; 10 mM HEPES, 1.2 mM $NaHCO_3$, 130 mM NaCl, 3 mM KCl, 2.5 mM $CaCl_2$, 0.6 mM $MgCl_2$, 20 mM glucose, and 20 mM sucrose at pH 7.4 and 290–300 mOsm). To monitor changes in intracellular $[Ca^{2+}]$ ($[Ca^{2+}]_i$), cells were loaded with either Fura 2-AM (HEK293 cells) or Fluo 4-AM (DRG neurons and mast cells) for 30 min in the dark at 37°C in CIB just prior to imaging. With Fura 2-AM, emission at 510 nm was monitored from excitation at both 340 nm and 380 nm. With Fluo 4-AM, emission at 520 nm was monitored from excitation at 488 nm. Cells were identified as responding if the intracellular $[Ca^{2+}]$ rose by either 50% compared to baseline or 50% compared to the $[Ca^{2+}]_i$ change assayed during addition of 50 mM KCl (neurons only). Damaged, detached, high-baseline, and motion-activated cells were excluded from analysis.

### HEK293 cells

In initial screens, HEK293 cells stably expressing the murine G-protein alpha-subunit $G_{\alpha15}$, a unique $G_\alpha$ protein that non-selectively couples a large variety of GPCRs to phospholipase C [30], were plated on poly-D-lysine-coated coverslips and transiently transfected with constructs encoding the MRGRPR of interest. 12–24 hr later, cells were loaded with the Fura 2-AM. Unless otherwise specified, compounds were perfused into the imaging chamber for approximately thirty seconds after a baseline period was established. Response was then monitored at 5 s intervals for an additional 60 s.

### DRG neurons

DRGs were incubated with Fluo-4 AM 24 hr after dissociation (native genotype) or 48 hr after dissociation (virally transduced). Unless otherwise noted, cells were imaged for 20 seconds to establish a baseline before compounds were added. After 30 s, a 2 min wash was applied before addition of another substances. At the end of every imaging trial, 50 mM KCl was added as a positive control. Cells included in calculating percentages all displayed at least a 50% increase in $[Ca^{2+}]_i$ compared to baseline upon addition of KCl. For ruthenium red inhibitor experiments, neurons were incubated with 10 µM ruthenium red for 5 min prior to imaging. Percentage activated was determined as earlier described.

### Mast cells

Mast cells were purified as described and plated onto glass coverslips coated with 30 mg/mL fibronectin and allowed to recover for 2 hr at 37°C. Cells were then loaded with Fluo-4 AM.

### $EC_{50}$ and $IC_{50}$ determinations

HEK293 cells stably expressing either MRGPRA1, MRGPRX4, and MRGRPC11 were seeded in poly-D-lysine-coated 96-well plates at 10,000 cells/well. Cells were loaded with Fura 2–AM, washed twice, and maintained in CIB. Haem metabolites were freshly dissolved in DMSO in dim light and then diluted into a buffer comprised of 20 mM Tris and 150 mM NaCl at pH 8.8. Potential changes in pH were evaluated prior to each experiment. $EC_{50}$ values were determined from dose-responses performed in triplicate, repeated 2–4 times. To determine potential antagonism by QWF against bilirubin, cells were treated with varying doses of QWF for 1 min in CIB prior to application of agonist.

## Murine peritoneal mast cell purification and calcium imaging

Adult male mice 8–12 weeks of age were sacrificed through $CO_2$ inhalation. A total of 25 mL of mast cell dissociation media (MCDM; HBSS with 3% fetal bovine serum and 10 mM HEPES, pH 7.2) was chilled on ice before being used to make two sequential peritoneal lavages. Lavages were combined and spun at 200 g. The pellet was re-suspended in 2 mL MCDM, layered over 4 ml of an isotonic 70% Percoll suspension (2.8 ml Percoll, 320 ml 10% HBSS, 40 ml 1 M HEPES, 830 ml MCDM), and spun for 20 min at 500 g at 4°C. Mast cells were recovered in the pellet. Mast cells were re-suspended in DMEM with 10% fetal bovine serum (FBS) and 25 ng/mL recombinant mouse stem cell factor (Sigma).

## Mouse peritoneal mast cell histamine release assay

Mast cells were purified as described and allowed to recover for 2 hr at 37°C. Cells were then seeded in 96-well plates coated with 20 mg/mL fibronectin at 300 cells/well. Plates were incubated at 37°C for 45 min before assay. For the assay, all compounds tested were diluted in CIB. Five minutes after compound addition, supernatant was aspirated and frozen at −80°C until histamine levels were determined with an HTRF histamine assay kit (Cisbio Assays) according to the manufacturer's instructions.

## Generation of cells stably expressing GFP-tagged MRGPRs

HEK293 stable cell lines expressing GFP-tagged MRGPRA3, MRGPRC11, MRGPRD, MRGPRX1, and MRGPRX2 were generated in previously described reports (*Liu et al., 2009*; *Liu et al., 2012*; *McNeil et al., 2015*). Briefly, plasmids containing the receptor of interest were transfected into HEK cells using Lipofectamine 3000. After 3 days, cells were then selected using 0.5 mg/mL G418. After 3 weeks, monoclonal colonies were established and each the highest expressing clones were

identified. For this study, *Mrgpra1* and *MRGPRX4* were inserted into pEGFP-N1 and transfected into HEK293 cells. MRGPR-positive cells were selected using 0.5 mg/mL G418 for three weeks, after which GFP-positive cells were sorted by FACS and monoclonally expanded. Two lines expressing similar levels of MRGPRA1 and MRGPRX4, as measured by GFP fluorescence, were selected for study.

## Microscale thermophoresis binding

Binding isotherms for MRGPRA1, MRGPRX4, and MRGRPC11 towards various ligands were determined by microscale thermophoresis with the NanoTemper monolith NT.115 instrument (*Duhr and Braun, 2006*). Ligands were pre-incubated with 10 µM of the GFP-tagged receptor of interest for 5 min at room temperature in binding buffer (20 mM Tris and 150 mM NaCl at pH 8.8). Receptors were crudely purified as a membrane fraction from cells stably expressing the receptor (*Vasavda et al., 2017*). Haem metabolites were freshly dissolved in 0.1 M NaOH in dim light and then diluted into assay buffer. Lyophilized BAM8-22 was dissolved in binding buffer. The pH of each ligand was evaluated prior to incubation with a receptor. Samples were loaded into NT.115 Hydrophobic-Treated Capillaries from NanoTemper. Microscale thermophoretic experiments were executed using 20% LED power and 15% MST power. $K_D$s were calculated using the law of mass action with data from three independent experiments. Binding between bilirubin and receptors was evaluated purely thermophoretically, whereas binding between BAM8-22 and MRGPRC11 was evaluated by T-Jump. Samples with dramatic deviations in initial fluorescence were excluded.

## [$^{35}$S]GTP$\gamma$S binding

MRGPR activation was determined by measuring binding of a radiolabelled and non-hydrolyzable form of GTP, [$^{35}$S]guanosine-5'-($\gamma$-thio)triphosphate ([$^{35}$S]GTP$\gamma$S) as previously described (*Vasavda et al., 2017*). Briefly, 10 µg of crude membrane fractions were diluted into 175 µL assay buffer (50 mM HEPES, 5 mM MgCl$_2$, 100 mM NaCl, 1 mM EDTA, 0.1% Triton 80) supplemented with 10 µM GDP and incubated at room temperature for 5 min. Membranes were then incubated an additional 1 min in a final volume of 199 µL assay buffer supplemented with 50 µM bilirubin. Samples were then brought to 200 µL with the addition of 10 nM [$^{35}$S]GTP$\gamma$S. Samples were incubated for 2 hr at 4°C with gentle agitation. The experiment was terminated by rapid filtration onto GF/B filters and washed three times with wash buffer (50 mM Tris-HCl, 5 mM MgCl$_2$, and 50 mM NaCl at pH 7.4). Filters were then immersed in scintillation cocktail and counted. Nonspecific binding was determined by competition with 10 µM unlabelled GTP$\gamma$S. GTP$\gamma$S binding assays were performed as two independent experiments, in triplicate.

## DRG dissociation and culture

DRG neurons from all spinal levels were collected in cold DH10 media (90% Dulbecco's modified Eagle's medium (DMEM)/F-12, 10% FBS, penicillin (100 U/mL), and streptomycin (100 µg/mL)). DRGs were digested with a dispase (5 mg/ml)/collagenase type I (1 mg/ml) enzyme mixture at 37°C for 45 min. After trituration, cells were spun at 300 g and re-suspended in DH10 before being plated on glass coverslips coated with poly-D-lysine (0.5 mg/ml) and laminin (10 µg/ml, Invitrogen). DRGs were cultured with DH10 supplemented with 50 ng/mL NGF at 37°C.

## DRG viral transduction

Lentiviruses encoding various cDNA for MRGPRs were generated using psPAX2 and pMD2.G. Virus was pelleted by centrifugation at 100,000 g for 4 hr, gently washed with twice with DH10 medium, and suspended in DH10. One day after DRG isolation and culture, DRGs were infected with lentivirus 24 hr overnight. The following morning, medium was completely replaced with fresh DH10 supplemented with 50 ng/mL NGF. 24 hr after infection, cells were processed for calcium imaging.

## DRG electrophysiology

DRG neurons from 3 to 5 week old mice were collected as described. After culture for 1–3 days, DRG neurons were transferred into a chamber with extracellular solution containing (in mM) 144 NaCl, 2.5 KCl, 2 CaCl2, 0.5 MgCl2, 5 HEPES, and 10 glucose, adjusted to pH 7.4 with NaOH. Whole-cell current-clamp recordings were performed at ~23°C using borosilicate capillary glass

electrodes (Sutter Instrument) with a tip resistance of 3–5 MΩ. Internal solution contained (in mM) 80 K-acetate, 30 KCl, 40 HEPES, and 1 CaCl2, adjusted to pH 7.4 with potassium hydroxide (KOH). Small-diameter neurons with diameter 15–25 µm were chosen for patch-clamp. Data were acquired using an Axopatch 700B Amplifier and Digidata 1322A Digitizer with pClamp9.2 software package (Axon Instrument). Chloroquine (CQ) in 1 mM was added by perfusion for 20 s, and bilirubin freshly made in 50 µM was added by pipette. Solutions containing 50 mM KCl were applied at the end of each cellular recording. Only neurons that could fire action potentials after adding KCl were regarded as healthy and appropriate for inclusion in data analysis.

## Immunohistochemistry

Adult mice (5–6 weeks old) were anesthetized by i.p. injection of chloral hydrate (20 µl/gram of 25 mg/ml solution) and perfused with 30 ml 0.1 M phosphate buffered saline (PBS) (pH 7.4, 4°C) followed with 30 ml of fixative (4% paraformaldehyde (vol/vol), 4°C). Skin, trigeminal ganglia, dorsal root ganglia, and spinal cord were dissected from the perfused mice. Tissues were post-fixed in fixative at 4° for 2 hr. Tissues were cryoprotected in 20% sucrose (wt/vol) for up to 8 hr followed by 30% sucrose for 24 hr and then sectioned (25 µm width) with a cryostat. The sections were dried at 37°C on slides for 1 hr and fixed with 4% paraformaldehyde at 21–23°C for 10 min. The slides were pre-incubated in blocking solution (10% normal goat serum (v/v), 0.2% Triton X-100 (v/v) in PBS, pH 7.4) for 1 hr at 21–23°C, then incubated overnight at 4°C with primary antibodies. Secondary antibody incubation was performed at 21–23°C for 2 hr. For primary antibodies, we used rabbit antibody to CGRP (T-4239, Peninsula, 1:1,000), rabbit antibody to GFP (A-11122, Molecular Probes, 1:1,000), and Substance P (rat monoclonal from Abcam, 1:250 dilution, M09205). For secondary antibodies, we used goat antibody to rabbit (A11011, Alexa 568 conjugated; A11008, Alexa 488 conjugated; Molecular Probes) and Invitrogen 547 (A-21247) for Substance P, all diluted 1:500 in blocking solution. To detect IB4 binding, sections were incubated with Griffonia simplicifolia isolectin GS-IB4 Alexa 568 from Invitrogen at 1:500 dilution (I-21412). Sections were washed three times with PBS and Fluoromount (Southern Biotech) was applied before cover slips were placed over section.

## In vivo DRG calcium imaging

Adult mice expressing Pirt-Cre and lox-stop-lox GCaMP6s were anesthetized by i.p. injection of chloral hydrate (20 µl/gram of 25 mg/ml solution). The back was shaved and disinfected with alcohol before application of ophthalmic ointment (Lacrilube; Allergen Pharmaceuticals). A dorsal laminectomy was performed below the lumbar enlargement (L5) targeting S1. During the procedure, care was taken to keep dura intact. A 2 cm incision was made at the lumbar enlargement. 0.1 mL of 1% lidocaine was injected into paravertebral muscles before dissection to expose L3–L5 vertebrae. Using rongeurs, the surface aspect of the L4 DRG transverse process was removed and the underlying DRG exposed. Mice were laid abdomen-down on a custom-designed microscope stage and the spinal column was secured at two sites using clamps. Images were acquired using a laser-scanning confocal microscope (Leica LSI microscope) equipped with a 53 0.5 NA macro dry objective and fast EM-CCD camera. Live images were acquired at 8 to 10 frames in frame-scan mode per 7–8 s, at depths of 0 to 70 mm below the dura with the DRG in the focal plane. Images were taken 30 min after peripheral stimulation of DRG via injection of vehicle or bilirubin by Hamilton syringe (5 µl) Throughout imaging, body temperature was maintained at $37 \pm 0.5$°C with a heating pad and rectal temperature monitoring. Anesthesia was maintained with 2% isoflourane and pure oxygen delivered through nosecone.

Raw image stacks were collected, deconvoluted, and imported into ImageJ (NIH). Optical planes from sequential time points were re-aligned and motion-corrected using the stackreg rigid-body cross-correlation-based image alignment plugin in ImageJ. Calcium signaling amplitudes were expressed as $F_t/F_0$ as a function of time. $F_0$ was defined as the average pixel intensity during the first two to six frames of each imaging experiment. All neurons that displayed a $> 0.25$ $F_0$ change from baseline were selected for further analysis. In subsequent analysis, neurons that displayed a $> 0.25$ $F_0$ change during either the baseline or the saline imaging periods were excluded from analysis.

## Mouse models of cholestasis and sample collection

1-naphthyl isothiocyanate (ANIT; Sigma) was solubilized in olive oil (Sigma). Animals were dosed with 25 mg/kg ANIT *per os* daily for five days. On day five, animals were acclimatized for itch behavior. On day six, animals were placed in test chambers and videotaped for one hour. The number of scratching bouts, defined as a continuous scratching movement with either hindpaw, was counted and binned in five minute intervals during the one hour period. After itch behavior was assessed, animals were administered pentobarbital (50 mg/kg, *i.p.*). Blood was collected by cardiac puncture and placed into heparinized tubes (BD Biosciences). After centrifugation, plasma was collected, aliquoted, and stored at −20°C until analysis. Bile acid levels were assessed by a fluorometric kit from Cell Biolabs. When applicable, mice were then proceeded for histology. For QWF antagonism of cholestatic itch, Day 5 ANIT-treated animals were dosed with either 1 mg/kg QWF dissolved in PBS or PBS vehicle *i.v.* via tail vein injection approximately 10 min before behavioral assessment of spontaneous itch. The dose was chosen based on previous studies (*Azimi et al., 2017*) as well as published pK data indicating stability in plasma ($t_{1/2}$ = 70 min).

## Skin bilirubin extraction

The skin of mice were exposed by applying a hair removal cream for 5 min, after which the skin was excised from the back and nape of mice and frozen at −80°C until processing. To extract skin bilirubin, skin was finely minced with a blade and then dounce homogenized at 4°C in 99% chloroform/1% glacial acetic acid (v/v). The organic layer was separated from any remaining tissue by centrifugation at 4°C at 16,000 g for 10 min. The organic layer was subsequently washed in 1% glacial acetic acid, then 0.2 M $NaHCO_3$, and then $H_2O$. The organic layer was evaporated with a speedvac until the pellet was dry, after which the pellet was resolubilized in buffer (50 mM Tris,150 mM NaCl, 1% TritonX-100, 5% glycerol at pH 7.4). The final pellet was resuspended as thoroughly as possible, but was unfortunately relatively insoluble. The calculated yield was 1.83 ± 0.2463 (SD) % and accordingly factored in to normalize skin bilirubin.

## UnaG purification

UnaG was expressed in pMAL-6P2-6xHIS in BL21(DE3) cells. Starter cultures were grown to saturation overnight in Luria Broth (LB) at 37°C. Starter cultures were then diluted 10-fold in LB and grown to an $OD_{600}$ of 0.3 at 37°C, after which cultures were moved to 18°C. At $OD_{600}$ of 0.6, protein expression was induced by the addition of 400 μM isopropyl β-D-1-thiogalactopyranoside for 16 hr at 18°C. Cells were harvested by centrifugation, resuspended in 15 ml of resuspension buffer (50 mM HEPES, 300 mM NaCl, 0.5 mM TCEP, 10% glycerol, 1 mM PMSF, 2.34 μM leupeptin, 1.45 μM pepstatin at pH 7.4) per liter of culture, and flash frozen in liquid nitrogen for storage at −80°C.

For purification, pellets were thawed on ice and sonicated to lyse. Lysate was clarified by centrifugation at 26,000 g at 4°C for 30 min and subsequently loaded onto an amylose column. Protein was eluted with 20 mM maltose in protease buffer (50 mM Tris, 150 mM NaCl, 0.5 mM TCEP, 10% glycerol, 0.01% TritonX-100 at pH 7.4), after which the MBP was were removed by the addition of Prescission Protease (GE Healthcare) for 16 hr at 4°C. UnaG was further purified by a second nickel-affinity step to remove the cleaved tag and Prescission Protease. Protein was further purified on a gel-filtration column (S-200, GE Healthcare) in 50 mM HEPES, 300 mM NaCl, 0.5 mM TCEP, 10% glycerol at pH 7.4. The purity of peak fractions was assessed by SDS-PAGE. The purified protein was concentrated using a 10 kDa MWCO filter (Amicon) and flash frozen in gel filtration buffer supplemented with 30% glycerol for storage at −80°C.

## Patient plasma and skin bilirubin quantification

To measure plasma bilirubin, patient plasma was diluted 1:20 in HBSS containing purified UnaG. To measure skin bilirubin, UnaG was added directly After a 10 min incubation at 25°C, bilirubin was quantified by interpolating from a standard curve. Samples with blood and hemolysis were excluded from analysis. Skin bilirubin was ormalized to the dry weight of the skin.

## Plasma bilirubin depletion

Plasma bilirubin was depleted either by selective oxidation by $FeCl_3$ to biliverdin IXα/biliverdin XIIIα or by immunoprecipitation. $FeCl_3$ was prepared as solution of 20% $FeCl_3$ in 0.1N HCl/methanol.

FeCl$_3$ was fluxed with plasma at a final concentration of 1.5% FeCl$_3$ at 37°C for 10 min. FeCl$_3$ is a mild oxidant but exhibits a redox potential that the oxidation of bilirubin to biliverdin (*Dolphin, 1978*). Bilirubin was also immunoprecipated by incubating plasma with 5 µg of either normal rabbit IgG or anti-bilirubin antibody (generated as previously described (*Doré et al., 1999*)) coupled to protein A/G beads for 1 hr at 25°C. To quantify bilirubin depletion, bilirubin was extracted from samples with 100% methanol and subjected to HPLC and UV-visible spectroscopic analysis. Absorbance was adjusted to a baseline of 0 OD, and bilirubin was quantified by integrating the area under the chromatographic peak.

## High pressure liquid chromatography (HPLC)

Plasma bilirubin was detected by HPLC using an analytical LC-18 column, 25 cm $\times$4.6 mm (Xterra, Waters Corporation). Bilirubin was eluted with gradients of mobile phases: 0.1 M ammonium acetate in 60% methanol/40% water (v/v) (pH 5.2) (Solvent A) and 100% methanol (Solvent B). Bilirubin was eluted as follows: 0 to 14 min: linear gradient from 100% A to 100% B; 14 to 19 min: linear gradient from 100% A to 100% B; 19–24 min: isocratic elution at 100% A. Bilirubin exhibited a retention time of approximately 14–15 min and was detected by measuring absorbance at 450 nm. The peak corresponding to plasma bilirubin was confirmed with the addition of 10 µM bilirubin to the sample as an internal standard.

## Data analysis

Group data were expressed as mean ±SEM unless otherwise noted. Two-tailed unpaired Student's t-tests, Fisher's exact test, and Chi-squared tests were used to determine significance in statistical comparisons, and differences were considered significant at $p<0.05$. Statistical power analysis was used to justify sample size, and variance was determined to be similar among all treatment groups as determined by F test. No samples or animals subjected to successful procedures and/or treatments were excluded from analysis. All behavior experiments were designed in a blocked manner with consideration for both genotype and treatment.

## Acknowledgements

We thank Dr. Cynthia Wolberger, Dr. Patrick Lombardi, Ms. Adele Snowman, Ms. Nadine Forbes, Mr. Chip Hawkins, and Dr. Xintong Dong for assistance with experiments. We would like to acknowledge the JHU Mouse Mutagenesis Core (P30NS50274) for the development of the A1 KO mouse. This work was supported by NIH Grants R01NS054791 and R01AI135186 (to XD) and MH18501 (to SHS).

## Additional information

### Competing interests

James Meixiong: is a consultant for Escient Pharmaceuticals a company focused on developing small molecule inhibitors for MRGPRs. Xinzhong Dong: has a financial interest in Escient Pharmaceuticals a company focused on developing small molecule inhibitors for MRGPRs. The other authors declare that no competing interests exist.

### Funding

| Funder | Grant reference number | Author |
| --- | --- | --- |
| National Institute of Mental Health | MH18501 | Solomon H Snyder |
| National Institute of Neurological Disorders and Stroke | R01NS054791 | Xinzhong Dong |
| Howard Hughes Medical Institute | | Xinzhong Dong |
| National Institute of Allergy and Infectious Diseases | R01AI135186 | Xinzhong Dong |

The funders had no role in study design, data collection and interpretation, or the decision to submit the work for publication.

## Author contributions
James Meixiong, Conceptualization, Data curation, Formal analysis, Validation, Investigation, Visualization, Methodology, Writing—original draft, Writing—review and editing; Chirag Vasavda, Data curation, Formal analysis, Validation, Investigation, Visualization, Methodology, Writing—original draft, Writing—review and editing; Dustin Green, Lijun Qi, Data curation, Investigation, Writing—review and editing; Qin Zheng, Data curation, Formal analysis, Investigation, Writing—review and editing; Shawn G Kwatra, James P Hamilton, Resources, Data curation, Writing—review and editing; Solomon H Snyder, Resources, Formal analysis, Supervision, Funding acquisition, Investigation, Project administration, Writing—review and editing; Xinzhong Dong, Resources, Formal analysis, Supervision, Funding acquisition, Validation, Investigation, Visualization, Methodology, Writing—original draft, Project administration, Writing—review and editing

## Author ORCIDs
James Meixiong (iD) http://orcid.org/0000-0001-6776-3975
Chirag Vasavda (iD) http://orcid.org/0000-0003-4558-4698
Xinzhong Dong (iD) http://orcid.org/0000-0002-9750-7718

## Ethics
Animal experimentation: All experiments were performed in accordance with protocols approved by the Animal Care and Use Committee at the Johns Hopkins University School of Medicine. All animals were handled according to approved institutional animal care and use committee (IACUC) protocols (MO16M40) of Johns Hopkins University.

## Decision letter and Author response
Decision letter https://doi.org/10.7554/eLife.44116.026
Author response https://doi.org/10.7554/eLife.44116.027

## Additional files
### Supplementary files
• Transparent reporting form
DOI: https://doi.org/10.7554/eLife.44116.024

### Data availability
All data generated or analyzed during this study are included in the manuscript. Source data for main figures 1-7 have been provided.

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
