## [Decision Letter]

Thank you for submitting your article "Identification of a bilirubin receptor that mediates a component of cholestatic itch" for consideration by *eLife*. Your article has been reviewed by three peer reviewers, including David Ginty as the Reviewing Editor, and the evaluation has been overseen by Catherine Dulac as the Senior Editor. The following individuals involved in review of your submission have agreed to reveal their identity: Mark Hoon (Reviewer #2); Ethan Lerner (Reviewer #3).

The reviewers have discussed the reviews with one another and the Reviewing Editor has drafted this decision to help you prepare a revised submission.

Summary:

This is an interesting, well-performed study that implicates bilirubin and the Mrgpr family members MRGPRA1 and human MRGPRX4 expressed on DRG and trigeminal sensory neurons as mediators of cholestatic itch. There is a lot to like about this study, including the finding that bilirubin promotes itch in mice, that it binds to and activates MRGPRA1 and human MRGPRX4, and using previously described and new knockouts these receptors mediate a component of cholestatic itch, including that evoked through transfer via human plasma. The reviewers are in favor of publication of this study in *eLife*. A few points for the authors to address/consider are noted below.

Requested revisions:

1) Does the injection of bilirubin into human subjects evoke itch? A positive result would markedly enhance the significance of this manuscript while a negative result would raise concerns. While not required for acceptance, this experiment would enhance the study.

2) Are there positive controls for the negative findings in the receptor activation assays? Are all other Mrgpr family members expressed and functional in the HEK293 cell calcium response assay? Please discuss this point.

3) There are only four figures and a lot of figure supplements. While not required, the authors may consider some re-organization of the figures.

4) In the summary paragraph, it is stated that MrgprA1 has not been characterized. That is not accurate. MrgprA1 has been characterized with respect to amino acid residues required for ligand binding and receptor activation, ligands including substance P and 48/80, and function, i.e., itch. In fact, MrgprA1 is functionally quite similar to MRGPRX2. Please discuss this point.

5) The EC_50_ for bilirubin on MrgprA1 and MRGPRX4 was determined to be 145.9 and 61.9 μM respectively. A comment could be added to the Discussion that this modest EC_50_ may contribute to why a markedly elevated bilirubin is needed to cause itch.

6) Previous work from this group has indicated that MrgprA1 may not be functionally relevant in adult mice. Are you now suggesting that MrgprA1 may be functionally relevant in adult mice? Do essentially all MrgprA3 neurons express MrgprA1? This topic is relevant to the sensory neuroscience community as the role of MrgprA1 in itch has been questioned.

7) It is stated that LPA does not activate MrgprA1 and that there is no difference between LPA-induced itch in WT versus MrgprA1^-/-^ mice. A close look at the data indicates that MrgprA1^-/-^ mice may scratch less than WT. There are many forms of LPA. We have data that certain LPAs activate MrgprA1. I am not suggesting more experiments, rather that the wording be less dogmatic and leave somewhat open the question of LPA interaction with Mrgprs.

8) Please acknowledge that neonates can have high levels of bilirubin in their skin but not itch, leaving open the discussion to possible mechanisms.

9) In the Materials and methods, it is noted that bilirubin must be handled with care and used rapidly. How was this stability concern addressed with respect to the human samples?

10) In Figure 1—figure supplement 2, there is a graphical depiction of suggested homologies between mouse and human *Mrgpr*s. This depiction is too simplistic. It should include the endogenous neuropeptide substance P, not only non-endogenous pruritogens chloroquine, compound 48/80. I am not sure what to do about BAM8-22 as it is sort of mixed. A line between A1 and Χ2 should be added. Even if their predominant expression is on different cells, there is no reason that the match between mice and human would be expected to be parallel. In addition, QWF is an antagonist of A1 and Χ2. If it is not an antagonist of the interaction between bilirubin and X4, then this would provide further data that the homology between A1 and X4 is imperfect. Please discuss this.

11) Summary paragraph. MrgprA1 was characterized as a receptor for FMRF, quinine, and partially responds to chloroquine by Liu et al. (2009). Therefore, it is incorrect to say that *Mrgpra1* is an uncharacterized *Mrgpr*. Please modify this statement.

12) Results, eighth paragraph. The human *MrgprX* family of receptors may have functional similarities between species but have no structural homologs in rodents, this should be clarified by citing Zylka et al. (2003) and Solinski, Gudermann and Breit (2014).

13) Previously the "Cluster knockout" was named Mrgpr-clusterΔ^−/+^ in previous publications by this group, to prevent confusion, it would be better if the same name was used in this publication.

14) Results, twelfth paragraph and Discussion, first paragraph. It would be reasonable to cite Alemi et al. (2013).

15) Since there is no direct data implicating either bilirubin or MRGPRX4 in human itch, the title should be modified. Once suggestion is the addition of the word 'may' before 'mediate(s)'.

---

## [Author Response]

Requested revisions:1) Does the injection of bilirubin into human subjects evoke itch? A positive result would markedly enhance the significance of this manuscript while a negative result would raise concerns. While not required for acceptance, this experiment would enhance the study.

We agree that injecting bilirubin into humans would provide valuable insight and directly answer whether bilirubin elicits itch. We are currently collaborating on an IRB-approved project (HIC#: 12780) to test whether injecting bilirubin into humans evokes itch. Unfortunately, completion of this trial will require substantial time and resources, making it difficult to complete in a timely and comprehensive manner. At least one author, J.M., has volunteered in the aforementioned study and can report an intense, long-lasting itch in response to pathophysiologic levels of bilirubin.

2) Are there positive controls for the negative findings in the receptor activation assays? Are all other Mrgpr family members expressed and functional in the HEK293 cell calcium response assay? Please discuss this point.

Thank you for highlighting an important assumption in our experiments. To ensure we would observe a calcium response following a true ligand-receptor interaction, we expressed each receptor in HEK293 cells stably expressing the G-protein alpha-subunit G_α15_, a Gα protein that couples GPCRs to intracellular calcium stores via phospholipase C (PLC) (Results, sixth paragraph). For Mrgprs with established ligands (MRGPRX1, Χ2, A1, A3, B2, C11, D), functionality and activity was validated by treatment with the established ligand after applying bilirubin (Figure 2—figure supplement 1). For Mrgprs without established ligands, we assayed for cellular expression of each receptor by qPCR.

3) There are only four figures and a lot of figure supplements. While not required, the authors may consider some re-organization of the figures.

We agree that the figures should be re-organized to improve clarity and readability. Accordingly, we have split several main figures into multiple figures and reduced the number of figure supplements by including supplemental data into the new main figures.

4) In the summary paragraph, it is stated that MrgprA1 has not been characterized. That is not accurate. MrgprA1 has been characterized with respect to amino acid residues required for ligand binding and receptor activation, ligands including substance P and 48/80, and function, i.e., itch. In fact, MrgprA1 is functionally quite similar to MRGPRX2. Please discuss this point.

We sincerely apologize for the incorrect description of MrgprA1 as “uncharacterized.” We have modified the statement to read “Bilirubin binds and activates two Mrgprs, mouse MRGPRA1 and human MRGPRX4” and we have also cited the relevant studies (Azimi et al., 2016, 2017).

5) The EC_50_ for bilirubin on MrgprA1 and MRGPRX4 was determined to be 145.9 and 61.9 μM respectively. A comment could be added to the Discussion that this modest EC_50_ may contribute to why a markedly elevated bilirubin is needed to cause itch.

We agree that the affinity of the receptors for bilirubin may explain why healthy individuals do not itch in response to normal levels of plasma bilirubin. We added the following comment to the Discussion: “The K_D_ of bilirubin towards MRGPRA1 and MRGPRX4 suggests that bilirubin likely only interacts and activates these receptors in individuals with markedly elevated bilirubin and not in healthy people.”

6) Previous work from this group has indicated that MrgprA1 may not be functionally relevant in adult mice. Are you now suggesting that MrgprA1 may be functionally relevant in adult mice? Do essentially all MrgprA3 neurons express MrgprA1? This topic is relevant to the sensory neuroscience community as the role of MrgprA1 in itch has been questioned.

We agree that the functionality and expression of MrgprA1 in adult mice is relevant to the sensory neuroscience community. Previous studies on MrgprA1 expression changes during development were determined by in situ *hybridization* which lacks sensitivity. Although adult mice may express less MrgprA1 than younger mice, we find that MrgprA1 is still expressed in the DRG, skin, trigeminal ganglion, and spinal cord in adult mice (Figure 3A-D). We have updated the text and figure legend to make it clear that these samples were taken from adult WT mice. Adult mice also scratch after treatment with the MrgprA1 agonist FMRF, further suggesting that MrgprA1 is also functional in adult mice. Critically, MrgprA1^-/-^ mice scratch significantly less than WT mice (Figure 2M). To emphasize this functional relevance of MrgprA1 in adult mice, we have amended the tenth paragraph of the Results.

7) It is stated that LPA does not activate MrgprA1 and that there is no difference between LPA-induced itch in WT versus MrgprA1^-/-^ mice. A close look at the data indicates that MrgprA1^-/-^ mice may scratch less than WT. There are many forms of LPA. We have data that certain LPAs activate MrgprA1. I am not suggesting more experiments, rather that the wording be less dogmatic and leave somewhat open the question of LPA interaction with Mrgprs.

We apologize for making the dogmatic claim that all LPAs do not activate MRGPRs. We have amended the text as follows: “Mrgprs are promiscuous receptors. It should be noted that there remain multiple bile acids, LPA molecules, and opiates which remain untested and could be agonists against Mrgprs.”

8) Please acknowledge that neonates can have high levels of bilirubin in their skin but not itch, leaving open the discussion to possible mechanisms.

We apologize for overlooking neonatal cutaneous jaundice in our Discussion and have amended the second paragraph of the Discussion to further emphasis bilirubin’s contextual and complex pruritic activity.

9) In the Materials and methods, it is noted that bilirubin must be handled with care and used rapidly. How was this stability concern addressed with respect to the human samples?

Thank you for highlighting our oversight. We have amended the Materials and methods section to indicate that plasma samples were maintained and manipulated in the dark to prevent bilirubin photolysis. We also reiterate in this section that plasma bilirubin was evaluated and quantified by HPLC. “To maintain the integrity of bilirubin in human plasma samples, samples were stored at -80°C until use. […] Plasma bilirubin was evaluated by HPLC as described above.”

10) In Figure 1—figure supplement 2, there is a graphical depiction of suggested homologies between mouse and human Mrgprs. This depiction is too simplistic. It should include the endogenous neuropeptide substance P, not only non-endogenous pruritogens chloroquine, compound 48/80. I am not sure what to do about BAM8-22 as it is sort of mixed. A line between A1 and Χ2 should be added. Even if their predominant expression is on different cells, there is no reason that the match between mice and human would be expected to be parallel. In addition, QWF is an antagonist of A1 and Χ2. If it is not an antagonist of the interaction between bilirubin and X4, then this would provide further data that the homology between A1 and X4 is imperfect. Please discuss this.

We apologize for over-simplifying and overlooking the diverse functional overlap and homology amongst the mouse and human Mrgprs. As the Mrgprs are particularly promiscuous and functionally diverse, we eliminated the graphic in order to avoid incorrectly suggesting rigid or universal human/mouse homology between receptors.

11) Summary paragraph. MrgprA1 was characterized as a receptor for FMRF, quinine, and partially responds to chloroquine by Liu et al. (2009). Therefore, it is incorrect to say that Mrgpra1 is an uncharacterized Mrgpr. Please modify this statement.

We apologize for the incorrect portrayal of MrgprA1 as “uncharacterized”. We have modified the statement to read “Bilirubin binds and activates two Mrgprs, mouse MRGPRA1 and human MRGPRX4.”

12) Results, eighth paragraph. The human MrgprX family of receptors may have functional similarities between species but have no structural homologs in rodents, this should be clarified by citing Zylka et al. (2003), and Solinski, Gudermann and Breit (2014).

Thank you for identifying and highlighting important studies and findings that add useful context to our work. We amended the paragraph to begin reading “The human MRGPRX family of receptors has functional similarities between species but have no obvious structural homologs in rodents.” We also cited the relevant studies suggested.

13) Previously the "Cluster knockout" was named Mrgpr-clusterΔ−/+ in previous publications by this group, to prevent confusion, it would be better if the same name was used in this publication.

Thank you for identifying this inconsistency in our text. At the request of a mouse genetics nomenclature expert that the journal contacted, we have amended the text and figures by replacing "Cluster ^-/-^" with “Mrgpr-cluster KO”. We note in the fourth paragraph of the Results, that Mrgpr-cluster KO refers to the previously published Mrgpr-clusterΔ^-/-^ mice.

14) Results, twelfth paragraph and Discussion, first paragraph. It would be reasonable to cite Alemi et al. (2013).

We have amended the text to cite this study in the Results and Discussion.

15) Since there is no direct data implicating either bilirubin or MRGPRX4 in human itch, the title should be modified. Once suggestion is the addition of the word 'may' before 'mediate(s)'.

We agree that our data are limited to mouse models of human itch and do not directly demonstrate that bilirubin plays a role in human cholestatic itch. Accordingly, we have amended our title to read “Identification of a bilirubin receptor that *may* mediate a component of cholestatic itch”.